# Observation of a reversal of rotation in a sunspot during a solar flare

Yi Bi[1,2], Yunchun Jiang[1], Jiayan Yang[1], Junchao Hong[1], Haidong Li[1,2], Bo Yang[1] & Zhe Xu[1,2]

The abrupt motion of the photospheric flux during a solar flare is thought to be a back reaction caused by the coronal field reconfiguration. However, the type of motion pattern and the physical mechanism responsible for the back reaction has been uncertain. Here we show that the direction of a sunspot's rotation is reversed during an X1.6 flare using observations from the Helioseismic and Magnetic Imager. A magnetic field extrapolation model shows that the corresponding coronal magnetic field shrinks with increasing magnetic twist density. This suggests that the abrupt reversal of rotation in the sunspot may be driven by a Lorentz torque that is produced by the gradient of twist density from the solar corona to the solar interior. These results support the view that the abrupt reversal in the rotation of the sunspot is a dynamic process responding to shrinkage of the coronal magnetic field during the flare.

[1] Yunnan Observatories, Chinese Academy of Sciences, Kunming, Yunnan 650216, China. [2] University of Chinese Academy of Sciences, Beijing 100049, China. Correspondence and requests for materials should be addressed to Y.B. (email: biyi@ynao.ac.cn).

The back reactions of solar coronal reconfiguration on the solar photosphere have been a long-standing issue in solar physics[1–7]. In particular, rapid and permanent changes of the photospheric magnetic field has been found over duration of the flare. Consistent with the conjecture of a magnetic implosion[8], the line-of-sight component has been observed to decrease rapidly[3,6] while the horizontal component was reported to substantially increase[2,9,10]. Another striking back reaction of the flare is the abrupt motion of the photospheric flux[1]. It is worth noting that the motion of magnetic flux could lead to the transport of magnetic helicity across the photosphere. It has been reported that, in the course of some major flares, magnetic helicity was impulsively transported across the photosphere and that the helicity flux tended to have the sign opposite to that of the active region[4,11,12]. However, the type of motion pattern responsible for the impulsive helicity transport is still not understood. Specifically, some researchers (for example, ref. 13) have reported sudden, shear-relaxing motions in the course of the flares. These motions may be driven by a horizontal Lorentz force that can be deduced from the abrupt changes of the photospheric magnetic field[14]. We note that such shear-relaxing motions may have a role in the impulsive variations of the helicity transport. Apart from the shear motion between the different flux elements, moreover, the internal spinning motion within an isolated flux element is another kind of tangential photospheric motion that could contribute to the helicity flux on the photosphere[15,16]. However, the role of the spinning motion in transferring helicity between the interior and the corona during a flare has been uncertain.

The most obvious cases in which internal spinning motion is directly observed are the so-called rotating sunspots, which can be regarded as a process that transfers magnetic helicity from the solar interior to the corona[17–20]. The clockwise (counter-clockwise) rotation of an upwardly directed magnetic flux tube transfers positive (negative) helicity into the corona[21]. The basic force responsible for such rotational motion is believed to be the Lorentz force[22–24].

Here, by means of a set of 45 s cadence full-disk continuum intensity images and 12 min cadence vector magnetograms taken by the Helioseismic and Magnetic Imager (HMI)[25] telescope on board the Solar Dynamic Observatory (SDO), we characterize the sudden reversal in the rotation of a part of a sunspot during an X1.6 flare. Moreover, the azimuthal component of the horizontal field in the sunspot is found to significantly increase over the course of the flare. On the basis of the change in the magnetic field, we obtain the change in the Lorentz torque exerted on the sunspot. Our estimation shows that the torque impulse is sufficient to reverse the rotation in the sunspot. Based on the results of a nonlinear force-free field (NLFFF) extrapolation model, we note that the magnetic field lines connecting the sunspot are shortened and become more twisted per unit length when the impulsive change occurs in the photospheric magnetic field. We suggest that the reversal in the rotation of the sunspot indicates that the increasing twist is transported downward across the photosphere.

## Results

**Overview of observations.** The flare SOL2014-09-10T17:45 (X1.6) lasted from 17:21 UT to 18:20 UT (Fig. 2a). The multi-wavelength extreme ultraviolet observations from the Atmospheric Imaging Assembly Telescope (AIA)[26] aboard the SDO show the flare ribbon in the 1,600 Å passband (Fig. 1a) and the post-flare loop in the 94 Å passband (Fig. 1b,c). The sunspot is located in NOAA active region 12,158, which is a mature simple active region. The photospheric intensity images (Fig. 1d–f) show

that a conspicuous light bridge splits the sunspot into the southern and northern parts. As shown in Fig. 1g, we used the Yet Another Feature Tracking Algorithm (YAFTA) package[15] to define and track the region labelled 'P'. This region encloses a flux concentration covering the southern part of the sunspot. As shown on the AIA images with 'P' outlined, the northern flare ribbon swept the southern part of the sunspot. The rotation of the sunspot can be deduced from the tangential component of the photospheric magnetic footpoint velocity $v$, which is obtained from the difference between two sets of 12 min cadence vector field data based on DAVE4VM[27]. Around the sunspot, the velocity field in 'P' shows a whirl pattern, which is counter-clockwise before the flare (Fig. 1h) but clockwise during it (Fig. 1).

**Magnetic helicity transport across the photosphere.** We obtain the magnetic helicity transport across the photosphere using the temporal sequence of vector field data and the associated DAVE4VM velocity field[21]. Negative helicity was continually injected into the active region during the analysed period of 72 h, but the impulsive change of the sign of the injected helicity occurred during the flare (Fig. 2b). The evolutionary characteristics of the velocity are well above the uncertainties that are estimated by the root mean square of 50 Monte Carlo experiments. In each experiment, we added Gaussian noise to three components of the vector magnetic field. The width of the Gaussian function for each pixel is taken as the noise level of the vector magnetic field, which is estimated based on the inversion code and provided by the HMI team[28]. The uncertainties in the vertical and tangential field are approximately 15 G and 50 G respectively, at each pixel.

The helicity flux can be decomposed into a shear term $\dot{H}_s$ and an emergence term $\dot{H}_e$, which yield the helicity flux contributed by the tangential and the vertical motion of the magnetic flux, respectively. In the studied period, the shear term dominated the emergence term and the impulsive helicity change was mainly attributable to the impulsive variation of the shear term during the flare (Fig. 2b). This suggests that no significant emerging magnetic flux occurred in the studied period and the injection of the impulsive helicity flux was mainly due to the shear motion of the magnetic flux during the course of the flare.

The shear term can thus be further decomposed in to a writhe term ($\dot{H}_{\mathrm{writhe}}$) and a spin term ($\dot{H}_{\mathrm{spin}}$), which refer to the contributions from the relative proper motions of photospheric flux elements about one another and from internal spinning motions within each element, respectively[15,16]. To assess the spin term attributable to the spinning motion of region 'P', the region outside 'P' is defined as the region 'O'. Based on Equation (5), the shear term in the active region can be expressed as $\dot{H}_{\mathrm{shear}} = \dot{H}_{\mathrm{spin}}^{P} + 2\dot{H}_{\mathrm{writhe}}^{PO} + \dot{H}_{\mathrm{spin}}^{O}$.

Figure 2c shows the temporal profiles of $\dot{H}_{\mathrm{spin}}^{P}$ and $\dot{H}_{\mathrm{writhe}}^{PO}$ as magenta and grey curves, respectively. Both the terms changed their signs during the flare and the change in $\dot{H}_{\mathrm{spin}}^{P}$ was higher than that of $\dot{H}_{\mathrm{writhe}}^{PO}$. Because no significant translational motions were detected in the centroids of either 'P' or 'O' in the course of the flare, the slight change in $\dot{H}_{\mathrm{writhe}}^{PO}$ during the flare indicates that the DAVE4VM velocities in 'P' do not accurately satisfy the conditions for the velocity field of a rigid body. As indicated by the dark curve in Fig. 3g, however, $\dot{H}_{\mathrm{spin}}^{O}$ was always negative and was nearly time-invariant. Accordingly, we may conclude that although the braiding motion of the flux surrounding 'P' continually contributed significant negative helicity, the reversal in the rotational motion of 'P' during the flare was the primary contributor to the abrupt change of helicity.

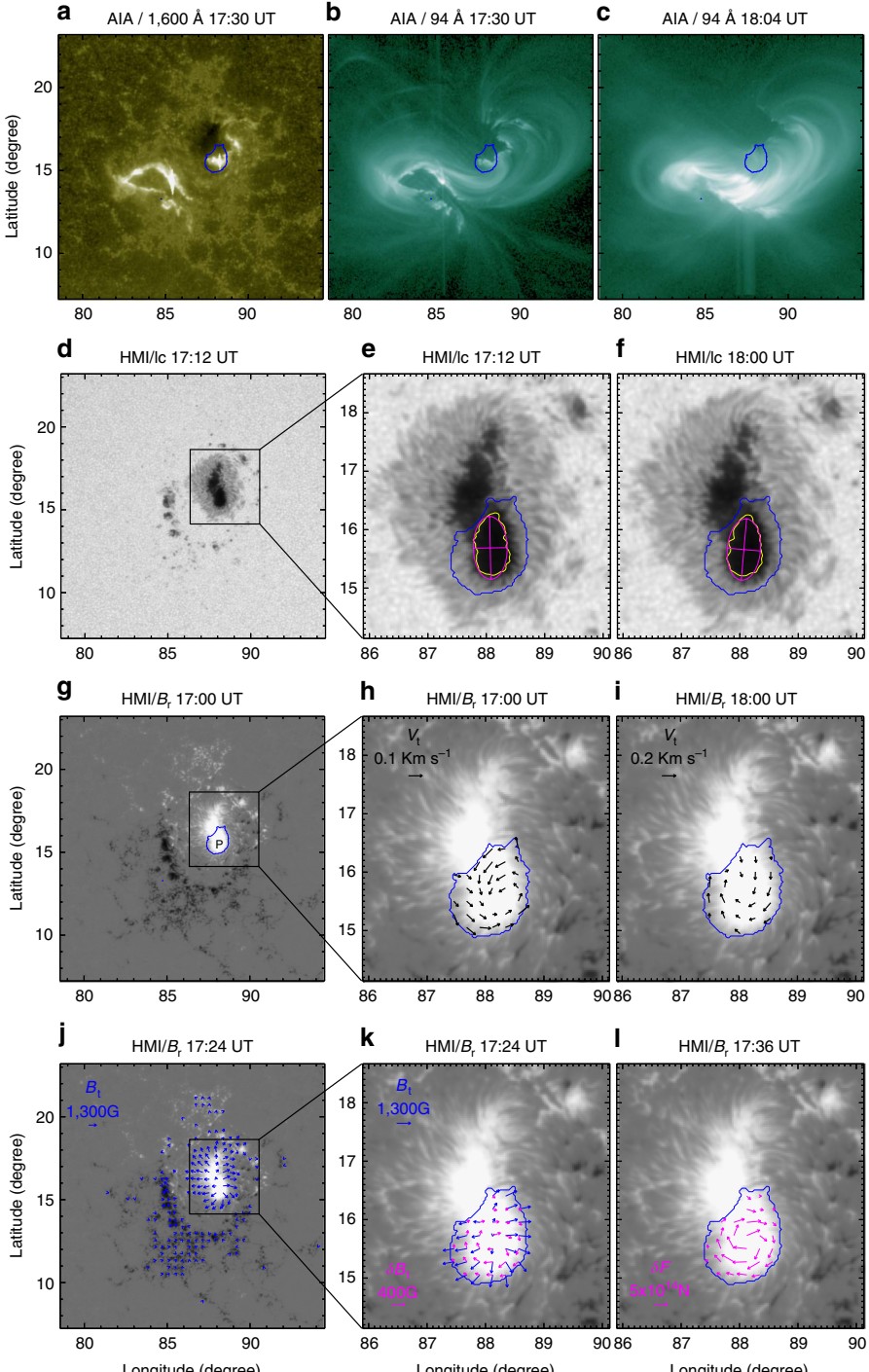

**Figure 1 | Overview of the flare and the rotational motion of the southern part of the sunspot observed by SDO/AIA and SDO/HMI.** (**a–c**) Comparison of AIA images at 1,600 and 94 Å showing the flare ribbon in the chromosphere and flaring loops in the corona, respectively. (**d**) SDO/HMI intensity images showing the whole active region in the left panel. (**e,f**) Close-up views of the intensity images covering the region of the sunspot. The yellow curves refer to the 13,000 data number contour level on the intensity image, which approximately outlines the umbral–penumbral boundary of the southern part of the sunspot. The magenta ellipses represent the best-fit ellipses that are obtained by applying the fit ellipse.pro procedure on the contour line. The fit ellipse.pro procedure is written in IDL and provided in the Coyote IDL programming package. The major and minor axes of the ellipses are also coloured magenta. (**g**) The background greyscale image shows the vertical field, with the positive field in white and the negative in black. It is scaled to $1,800.0 \, \mathrm{Mx \, cm^{-2}}$. (**h,i**) Close-up views of the image in **g** are superimposed with tangential velocity vectors (black arrows) inferred from the DAVE4VM technique. (**j–l**) As in **g**, the background greyscale shows the vertical field. In **j** and **k**, the superimposed blue arrows refer to the tangential field vectors. The magenta arrows in **k** and **l** refer to changes in the tangential field and changes in the tangential Lorentz force, respectively. The blue curve in each panel outlines the region that is labelled by 'P', which corresponds to a flux concentration (in images of the total magnetic field $|B|$) identified with the Yet Another Feature Tracking Algorithm (YAFTA) package. The tracked region 'P' appears to cover the southern part of the sunspot in the intensity image shown in **e** and **f**.

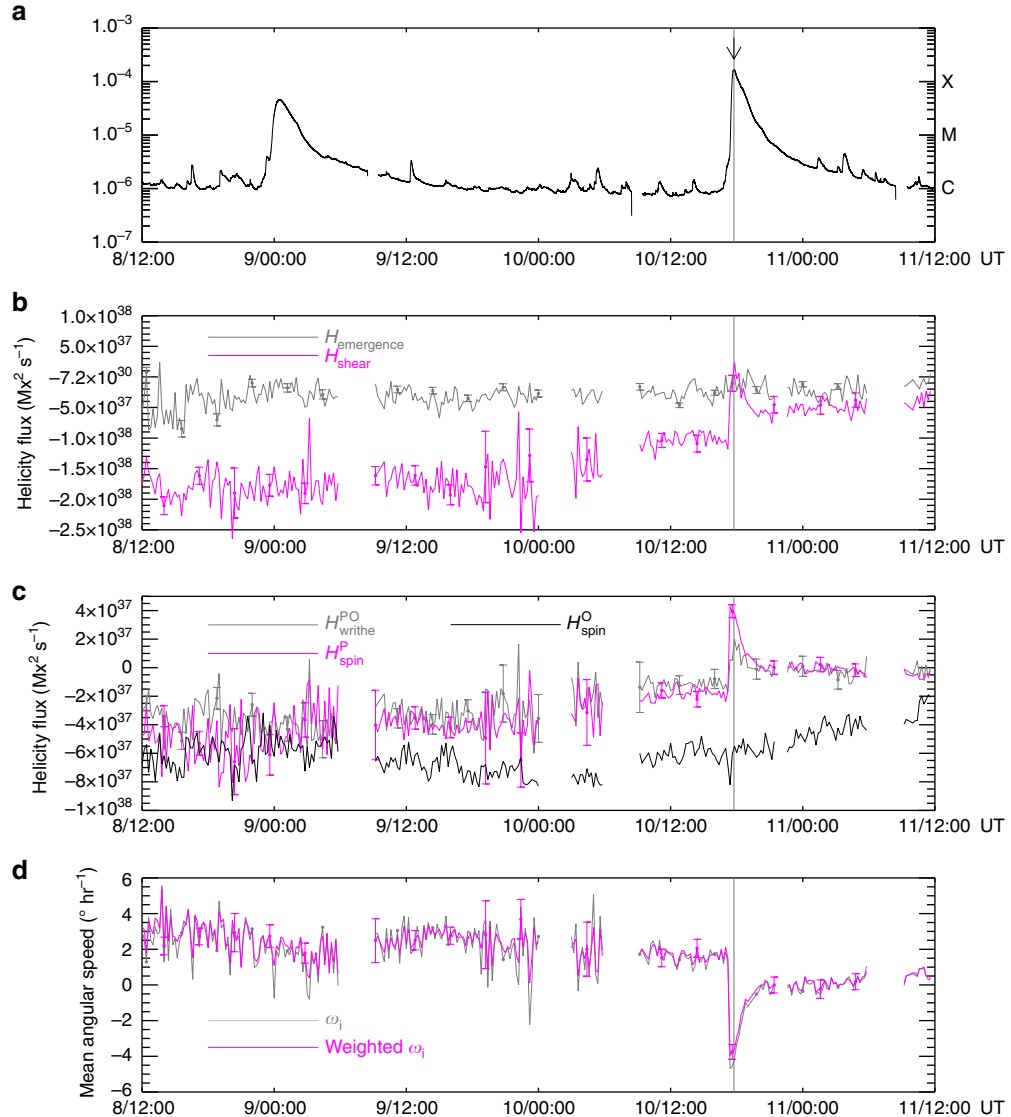

**Figure 2 | Helicity flux across the photosphere.** (**a**) The GOES 1–8 Å light curve. (**b**) Temporal profiles of the magnetic helicity flux across the photosphere over the same field of view (FOV) as Fig. 1g. The magenta and grey curves represent the shear term $\dot{H}_{shear}$ and the emergence term $\dot{H}_{emergence}$ of the helicity flux, respectively. (**c**) Temporal profiles of helicity flux across the photosphere from spin term $\dot{H}_{spin}^{P}$ (magenta), spin term $\dot{H}_{spin}^{O}$ (grey) and the writhe term $\dot{H}_{writhe}^{PO}$ (black). Region outside 'P' and inside the FOV of Fig. 1g is defined as the region 'O'. (**d**) Temporal profile of the mean angular velocity of 'P'. The magenta and grey curves represent temporal profiles of the angular velocity which are obtained from Equation (9) and Equation (10), respectively. In each panel, the vertical grey bar indicates the flare peak time and the error bars represent 1 s.d.

**Mean angular velocity of the sunspot**. Based on Equation (9), the spin term of the magnetic helicity flux is proportional to a mean rotation velocity weighted by the magnetic flux[16]. The magenta line in Fig. 2d presents the temporal profile of the weighted rotational velocity of 'P'. It is obvious that in the 3-day period the pronounced reversal in the rotation occurred only during the flare SOL2014-09-10T17:45 and the rotation rate reached approximately $-4°\,h^{-1}$. The sunspot rotated at an approximately constant speed of $2°\,h^{-1}$ before the flare and the rotational velocity was close to zero after the flare[18]. The temporal profile of the mean angular velocity around the flare time can be seen in Fig. 3a.

**Rotation of the best-fit ellipse on the sunspot**. Impulsive clockwise rotation during the flare is also detected in the successive 45 s cadence HMI intensity images. Supplementary

Movie 1, created from the intensity images, shows that both the southeast and southwest edges of the sunspot move eastward and that the light bridge, located along the northeast edge of the southern part of the sunspot, moves westward. These are consistent with an impulsive clockwise rotation of the southern part of the sunspot. Because the southern part of the sunspot has an elliptical shape[29], we studied the rotation of the sunspot by tracking the variations of the orientation of the ellipses that best fit the sunspot on a series of intensity images. As shown in Fig. 1e,f, the overlaid ellipses are the best fits to the 13,000 data number contour level on the intensity images, which approximately outline the umbral–penumbral boundary of the sunspot.

After carefully examining the intensity images, we find that all of the contour lines at contour levels of data number varying between 13,000 and 17,000 are good matches to the umbral–penumbral boundary of the sunspot in the intensity images taken

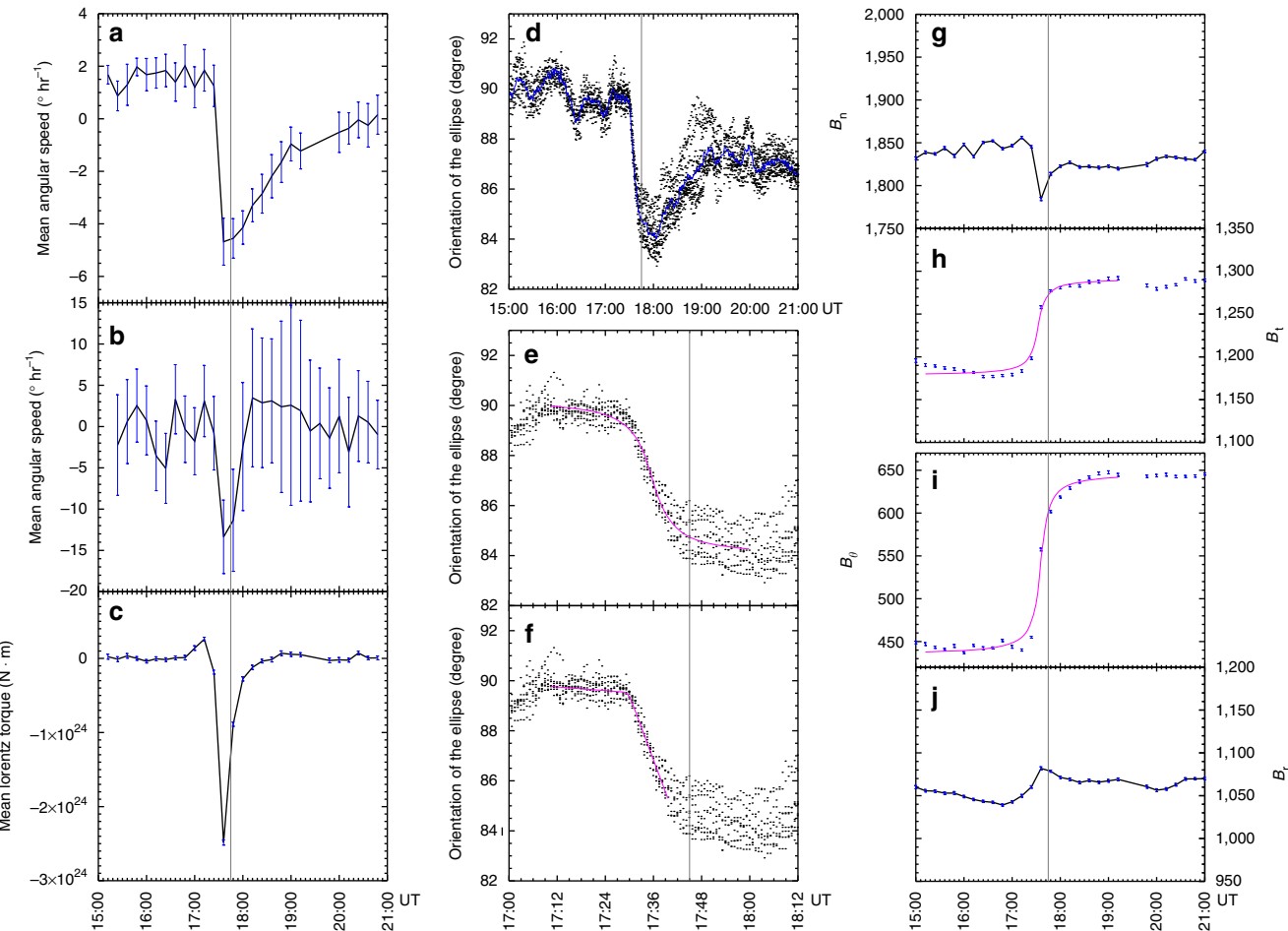

**Figure 3 | Temporal profiles of the kinematic parameters and the mean photospheric magnetic field of the sunspot.** (**a**) The temporal profile of the mean angular velocity of the region 'P' derived from the DAVE4VM velocity. (**b**) The temporal profile of the rotation rate of the ellipse obtained by performing a numerical differentiation of the 12 min average orientations of the best-fit ellipse. (**c**) The temporal profile of the change in the Lorentz torque exerted on region 'P'. (**d**–**f**) The points denote the orientations of the nine ellipses, each of which is the best fit to one of the contour levels of data number varying between 13,000 and 17,000. In **d**, the blue curve denotes the temporal profile of the average orientations of the nine ellipses. In **e** and **f**, the magenta curve represents the best fit of Equation (13) and Equation (1) to the average orientations, respectively. (**g**,**h**) The temporal profiles of the mean vertical field ($B_n$) strength and the mean tangential field ($B_t$) strength within region 'P'. (**i**,**j**) The temporal profiles of the mean of the azimuthal component ($B_\theta$) and the radial component ($B_r$) of $B_t$ within region 'P'. In **h** and **i**, the magenta curves represent the best fit of Equation (13) to the data. In each panel, the vertical grey bar indicates the flare peak time and the error bars represent 1 s.d.

around the time of the flare. Supplementary Movie 1 shows the evolution of ellipses fitting the nine different data number contour lines. The selected data numbers range from 13,000 to 17,000 and are spaced at regular intervals of 500. As shown in Supplementary Movie 1, all the fitting ellipses appear to undergo a sudden clockwise rotation during the flare, although the details of the contour lines are different. Therefore, it is possible that the measured change in ellipse orientation was caused by global rotation of the sunspot instead of movement or change in the umbral structure. In Fig. 3d, the points mark the orientations of the ellipses fitting different contour levels at each moment. It clearly shows that the data points taken at approximately 17:36 UT are rather close to each other, indicating that the ellipses fitting different contour levels rotate in an identical way in this period. Thus, it seems that the changes in the orientations of the ellipses exactly mark the rotation of the sunspot at approximately 17:36 UT.

Taking advantage of the 45 s cadence intensity images, we can further detail the abrupt rotation of the sunspot. The magenta curve overlaid in Fig. 3e denotes the fit of a step function (Equation 13) to the orientations of the fitted ellipses, which

shows that the sudden reversal of the sunspot occurred from 17:31 UT to 17:41 UT and the average angular velocity reached $-24 \pm 5^\circ\,h^{-1}$ for a period of 10 min. Moreover, the magenta curve in Fig. 3f indicates the best-fits of Equation (1) to the ellipse orientations around the angular acceleration phase, indicating that the spinning motion of the sunspot accelerated within $96 \pm 20$ s, from approximately 17:29 UT to approximately 17:31 UT.

To compare the rotation rate of the fitted ellipses with the mean angular velocity deduced from the 12 min cadence vector field, as shown in Fig. 3b, we obtain the rotation rate of ellipse by performing a numerical differentiation to the 12 min average orientations of the ellipse. Here, a $1\sigma$ error indicates the standard deviation of the numerical deviation. It can be seen that in the rapid rotation phase, the change in the rotation rate of the ellipse is well above the error. In this period, however, the rotation rate of the ellipse reached $-10 \pm 3^\circ\,h^{-1}$, which is larger than the average angular velocity of $-4^\circ\,h^{-1}$. It is worth noting that the former denotes the apparent footpoint motion of the umbra. The apparent photospheric motion is partly attributed to the normal plasma velocity[30]. However, the significant vertical flow cannot be detected from DAVE4VM. An alternative

interpretation for the discrepancy would be that the sunspot rotated with slightly inhomogeneous angular velocity and the umbral rotated faster. Moreover, the abrupt change in the ellipse fitting could not be completely attributed to a strictly proper bulk motion of the sunspot, but could be partly attributed to signatures of opposite changes in the sunspot structure at the penumbrae, for example, flattening of the penumbra at the southeast or steepening at the southwest.

**Photospheric magnetic field changes**. As shown in Fig. 3g,h, the mean vertical field within region 'P' diminished slightly, while the mean tangential field ($B_t$) within region 'P' increased from approximately 1,200 G to 1,300 G in 24 min. If we transform $B_t$ into polar coordinates ($r$, $\theta$) with $r = 0$ located at the centre of the region, the mean $B_t$ within region 'P' can be represented by a radial component ($B_r$) and an azimuthal component ($B_\theta$). During the flare, the changes in $B_r$ were weak (Fig. 3i) and the changes in $B_t$ were mainly attributed to the approximately 135 G increase in $B_\theta$ (Fig. 3j). Again, uncertainties were obtained by conducting Monte Carlo experiments.

**Photospheric Lorentz force changes**. Based on the changes in the photospheric vector field during the flare, we can estimate the corresponding change in the horizontal Lorentz force acting on the photosphere[31]. The force acted on the sunspot in the clockwise direction and produced a torque about the vertical axis in the downward direction (Fig. 1l), which is consistent with the clockwise rotation of the sunspot observed during the flare (right panel of Fig. 1i). With the centroid of the sunspot as the axis, we obtain the change of the axial torque during the flare to be approximately $2.5 \times 10^{24}$ Nm, one order of magnitude larger than that before and after the flare.

The azimuthal perturbation of the photospheric magnetic field generates a torsional Alfvén wave that propagates with the Alfvén speed of $V_A = B_z/(4\pi\rho)^{1/2}$. Considering that the magnetic field is approximately homogeneous between the photosphere and approximately 2 Mm below[32], we take $B_z$ to be the surface magnetic field strength. Moreover, we adopt the Model S[33] to estimate the density below the photosphere. Accordingly, the torsional Alfvén pulse could propagate to a depth of approximately 0.5 Mm below the photosphere within 2 min, during which the sunspot is detected to undergo an angular acceleration as discussed above. Thus, the Lorentz torque could act on the underlying magnetic flux to a depth of 0.5 Mm. The flux tube involved in the reverse rotation would then contain the mass on the order of $5 \times 10^{16}$ kg and have a moment of inertia on the order of $2 \times 10^{30}$ Nm$^2$, which is estimated by $I = \sum m_i(r_i - r_0)^2$, where $r_0$ is the centroid of region 'P'.

If the photospheric field changes impulsively such that the magnetic field below the photosphere does not have time to respond, the changes in the Lorentz force applied to the photosphere would cause a net rotational torque to be exerted on the sunspot and then produce an angular acceleration of the sunspot. For simplicity, we assume that the Lorentz torque $\delta T$ occurs within the time from $t = 0$ to $t = \delta t$ and the Lorentz torque impulse is related to the angular momentum by $\delta T \delta t = I\omega$, where $I$ is the moment of inertia and $\omega$ is the resulting angular velocity. It can then be estimated that the Lorentz axial torque of order $2.5 \times 10^{24}$ Nm could accelerate the sunspot to an angular velocity of order $30° \, \text{h}^{-1}$ during over the entire 2 min period. Thus, the estimation suggests that the Lorentz torque is sufficient to reverse the rotation of the sunspot during the course of the flare.

**Coronal magnetic extrapolation**. To shed light on the magnetic field structure of the corona, we construct the coronal magnetic field $B$ using a NLFFF extrapolation model[34–37]. The model is applied by using the photospheric field data as boundary conditions and assumes that the corona is static and free of Lorentz forces, such that $\nabla \times B = \alpha B$, where the force-free parameter $\alpha$ identifies how much current flows along each field line and is invariant along field lines. During the changeover period of the photospheric field, the mean photospheric $\alpha$ within region 'P' shows an impulsive enhancement. Consistently, the NLFFF modelled magnetic field lines that originate from region 'P' evolve towards higher values of $\alpha$, and the mean $\alpha$ increases $29 \pm 5\%$ within 24 min (Fig. 4d). The sudden change in the mean $\alpha$ is well above the errors, which is estimated by performing the extrapolation 50 times on the vector data with added Gaussian noise.

The NLFFF field lines become shorter as the photospheric magnetic field changes (Fig. 4a,b). The mean length of the field lines decreases $14 \pm 0.5\%$ within 24 min during the flare (Fig. 4e). However, the illustration from the NLFFF extrapolation is purely schematic without any reference to the reconnection process itself. Nevertheless, the models suggest a scenario in which the shortening of the field lines plays a role in increasing the $\alpha$ of the field lines, even if the change in the twist of the field is relatively minor. In addition, the twist of a magnetic field line is expressed by $T = \frac{1}{4\pi}\bar{\alpha}L$, where $L$ is the length of the field line and the force-free parameter $\alpha$ can be regarded as a local density of twist along the field line[38]. The mean twist of the field associated with the sunspot did not show a striking change with the occurrence of the flare. As $\alpha$ increased and the length decrease $d$, the mean magnetic twist was found to increase slightly during the flare, but the change was not significantly greater than the statistical error (Fig. 4f).

## Discussion

Using the magnetic field measurements made by HMI, we obtained the solid evidence of a sudden reversal of sunspot rotation direction and a rapid enhancement in the azimuthal component of horizontal magnetic field in the sunspot. The abrupt change in photospheric field indicates a Lorentz torque acting on the photosphere. This magnetic torque has the same direction as the rotation of the sunspot during the flare. Rapid changes in the photospheric field and motions at the photosphere during the flare are more likely driven by the flare than by convective flows because the photospheric convective turnover time at this length scale is much longer than the timescale of the changes. Therefore, the impulsive rotation of the sunspot and the rapid change in the magnetic field are more probably caused by a dynamic process responding to the coronal field reconfiguration during the flare. Moreover, the Lorentz torque could be sufficient to accelerate the reversal of rotation in the sunspot on a timescale of 2 min. It must be noted that the reaction of the field below the surface to the coronal torque is not taken into account in the calculation. There may be opposition from the interior to balance the change in the field on the photosphere. However, the change in the interior of the sunspot responding to the coronal field reconfiguration may lag behind the change in the photosphere.

It has been reported that an abrupt decrease in azimuthal field occurred in two sunspots in NOAA 11158 during an X2.2 flare[39] and the corresponding Lorentz force changes were consistent with the stepwise increased rate of sunspot rotation[7]. The authors[40] noted that the sunspots may connect the two ends of the flux rope associated with the flare, and suggested that the decrease in the horizontal field of the sunspots could be explained by the elimination from the flux rope of some of the twist

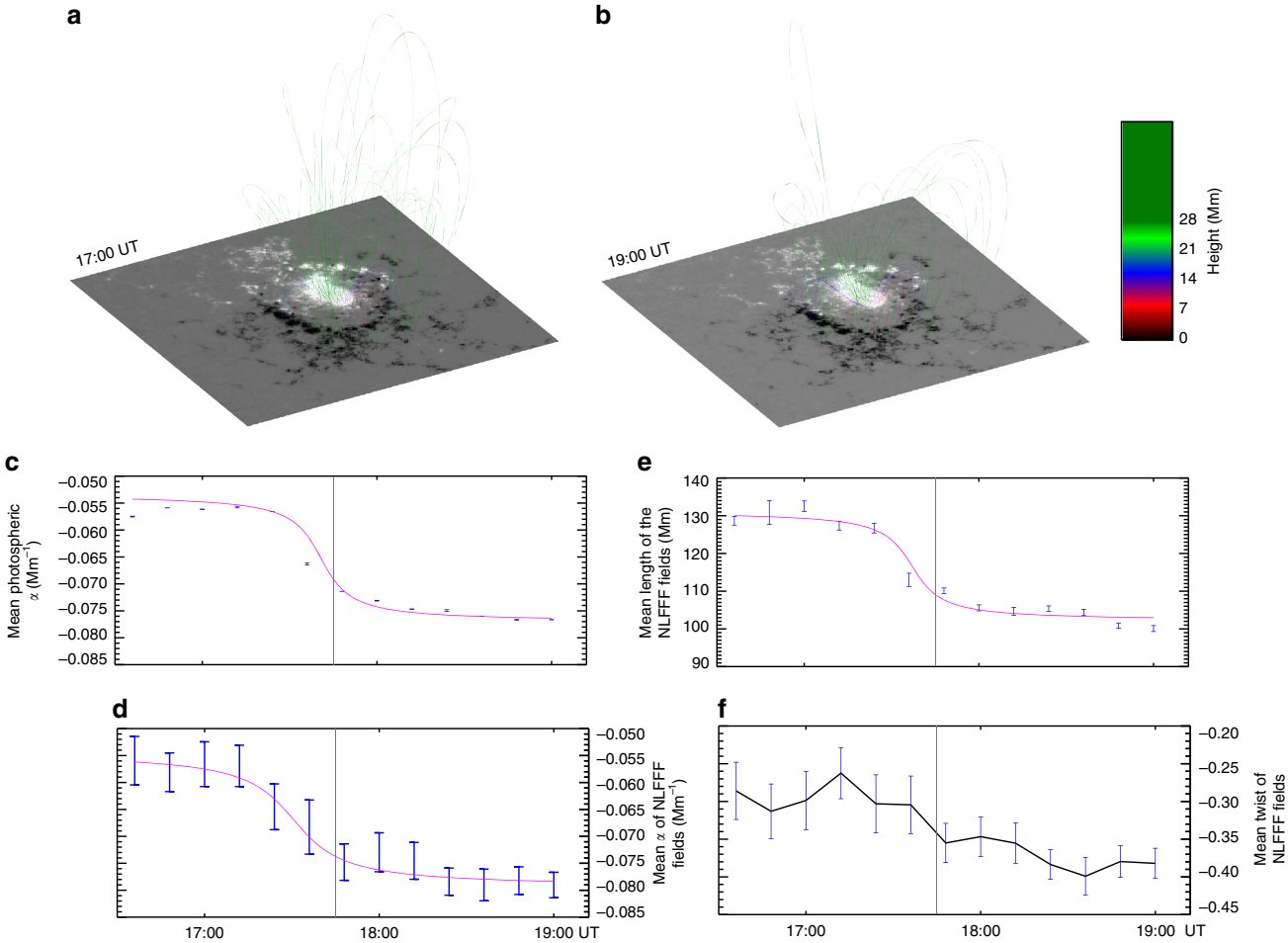

**Figure 4 | The results of the NLFFF extrapolation.** (**a**,**b**) Three-dimensional views of field lines traced from the region 'P' before and after the flare. Each field line is colour-coded according to the altitude of its apex. (**c**) The temporal profile of the mean photospheric α within region 'P'. (**d**–**f**) The temporal profile of the mean α, the mean length and the mean twist of the field lines traced from region 'P'. In **c**–**e**, each curve represents the fit of Equation (13) to the data. In **c**–**f**, the vertical grey bar indicates the flare peak time and the error bars represent 1 s.d.

component of the field[41,42]. In contrast to the previously studied case, the sunspot presented here was swept by the flare ribbon. Therefore, the physical mechanism evolving in the sunspot during the flare should be similar to that evolving in the magnetic flux along the flaring magnetic polarity inversion line (PIL), at which a significant increase in horizontal field and a relatively small change in the vertical field are often found. Such change in photospheric field implies that there are Lorentz-force changes acting on the photosphere, which trend to relax the shear of the field near the PIL. However, the enhancement of the horizontal field near the PIL is often accompanied by an increase in magnetic shear at the core-flaring region[43].

Outside of flares, torsional oscillations of some sunspots were noticed and the directions of rotation of those sunspots were found to be reversed periodically[44]. The authors reported the period of oscillations as approximately 3.8 days[45]. However, when we traced the rotation of the sunspot reported here using the DAVE4VM velocity field, we found that no torsional oscillations occurred in this sunspot in our studied period and the counter-clockwise rotation of this sunspot lasted for approximately two days before the flare. The rotation of the sunspots is thought to be driven by a Lorentz torque that is produced by the gradient of α along the flux tube from the solar interior to the atmosphere[22,46]. Such a gradient in α is suggested to result from low α magnitude in the corona due to the extreme expansion and stretching of the magnetic field lines after an initial stage of flux emergence. In the

reported event, the counter-clockwise rotation of the sunspot indicates that the twist was injected into the corona, and that the α in the corona should be lower than that in the interior before the flare. In this scenario, the reversal in rotation of the sunspot requires the α in the corona to be greater than that in the interior. Owing to the longer timescale of the convective flows in the interior, the change in the interior α should be much lower than in the corona during the period of the flare. Thus, it is more probable that a rapid enhancement in α occurred in the corona.

Consistently, the NLFFF model indicates that the magnetic field originating from the sunspot has a stepwise enhancement in the value of α during the flare. In the meantime, the length of the modelled field lines appears to significantly decrease. It has been reported that the stepwise enhancement in the horizontal magnetic field along the PIL was accompanied by the collapsing of the associated NLFFF modelled field[47–51]. Here, similarly, the shortening of the field as well as the increase in the horizontal field of the sunspot support the contraction of the magnetic field lines in an energy-releasing coronal transient event, as predicted by the conjecture of a magnetic implosion[8]. In addition, the field lines originated from the sunspot share the same footpoints with the observed post-flare loop system, indicating that the subsidence of the fields may have resulted from the shrinkage of the newly formed flare loops[52,53].

Because the force-free parameter α corresponds to the local density of the magnetic twist, the shortening of a twist field

should increase its $\alpha$ value. It is possible, then, that the reconnected field during the flare would have a higher value of $\alpha$ if the reconnected fields significantly shrink with considerable twist. In fact, the shear-flaring loops have been reported to appear in the early phase of the flare[54], implying a reconnection-driven transfer of magnetic shear from pre-flare to post-flare magnetic fields[55].

It has been reported that magnetic helicity was impulsively transported across the photosphere in the course of some major flares[4,11,12]. The authors further noted that the impulsive helicity flux tended to decrease the magnetic helicity of the entire active region. Here, the sudden reversal in the rotation of the sunspot supports this tendency. The rapid changes detected in the photospheric magnetic field are another back reaction of the coronal reconfiguration on the photosphere during the flare occurrence. The impulsive enhancement in the azimuthal component of the photospheric magnetic field suggests the contraction of the coronal magnetic field with increasing magnetic twist density. Accordingly, the impulsive magnetic helicity flux across the photosphere, as indicated by a sudden rotation of the sunspot, possibly results from the transportation of magnetic twist from the corona to the solar interior during the flare.

## Methods

**Observations and data processing.** The essential observational evidence of the rotation of the sunspot is provided by the HMI instrument on board the SDO. HMI takes full-disk continuum intensity images with a pixel size of $0''.5$ and 45 s cadence. The vector magnetic field data[56] provided by HMI is available every 12 min, computed using the Very Fast Inversion of the Stokes Vector code[57]. The remaining 180° azimuth ambiguity is resolved with the Minimum Energy code[58,59]. The dynamic range limitations of the instrument do not become significant until nearly 3,000 G (ref. 60). Taking advantage of the HMI field data, here, we investigated the evolution of the photospheric magnetic field in a sunspot that has a mean magnetic field strength of $\sim 2,200$ G. The associated chromospheric and coronal structures are examined using extreme ultraviolet data from the AIA on board the SDO. The AIA takes full-disk multi-wavelength images with a pixel size of $0''.6$ and a cadence of 12 s. All the data were remapped to a Lambert Cylindrical Equal-Area projection and then transformed into standard heliographic spherical coordinates.

**The best-fit ellipse on the sunspot region.** The rotational motion of the sunspot is estimated based on the evolution of the shape of the southern part of the sunspot in the successive HMI intensity images. Because the southern part of the sunspot is elliptical, its orientation is measured from the major axis of the best-fit ellipse to the Equator of the Sun[29]. Here, the parameters of the best-fit ellipse are obtained by applying the fitellipse.pro procedure on the identified boundary of the sunspot. The fitellipse.pro procedure is written in IDL and provided in the Coyote IDL programming package.

To estimate the timescale of angular acceleration of the best-fit ellipse, we fit these elliptical orientations with a function of the form[61]

$$f(t) = f_0 + \frac{1}{2}(v_\mathrm{f} + v_0)(t - t_0) + \frac{1}{2}(v_\mathrm{f} - v_0)\tau \ln[\cosh(\frac{t - t_0}{\tau})] \quad (1)$$

Here, the acceleration occurs mainly on the interval $[t_0 - \tau, t_0 + \tau]$, and $2\tau$ is the timescale of the acceleration. The uncertainty of the timescale is estimated by the standard deviation of the best-fit $\tau$

**Methods for helicity fluxes.** The rate of helicity transport across the photosphere is expressed by[21]

$$\frac{\mathrm{d}H}{\mathrm{d}t} = \underbrace{2 \int_S (\mathbf{A}_\mathrm{p} \cdot \mathbf{B}_\mathrm{t}) V_{\perp\mathrm{n}} \mathrm{d}S}_{\dot{H}_\mathrm{e}} - \underbrace{2 \int_S (\mathbf{A}_\mathrm{p} \cdot \mathbf{V}_{\perp\mathrm{t}}) B_\mathrm{n} \mathrm{d}S}_{\dot{H}_\mathrm{s}} \quad (2)$$

where $\mathbf{A}_\mathrm{p}$ denotes the vector potential of the potential field, $\mathbf{B}_\mathrm{t}$ and $B_\mathrm{n}$ are the tangential and normal magnetic fields and $\mathbf{V}_{\perp\mathrm{t}}$ (resp. $V_{\perp\mathrm{n}}$) indicate the tangential (resp. normal) components of $V_\perp$, the velocity perpendicular to the magnetic field. The first term arises from twisted magnetic flux tubes emerging from the solar interior into the corona (emergence term $\dot{H}_\mathrm{e}$ hereafter), while the second term is generated by shearing and braiding the field lines by tangential motions on the solar surface (shear term $\dot{H}_\mathrm{s}$ hereafter). If we consider the photospheric surface S as

a plane, then this equation can be rewritten as[62,63]

$$\dot{\mathbf{H}}_\mathrm{e} = \frac{1}{2\pi} \int_S \int_{S'} \mathrm{d}^2x \mathrm{d}^2x' \hat{\mathbf{n}} \cdot \frac{x - x'}{|\,x - x'\,|^2}$$
$$\times \{\mathbf{B}_\mathrm{t}(x)V_{\perp\mathrm{n}}(x)B_\mathrm{n}(x') - \mathbf{B}_\mathrm{t}(x')V_{\perp\mathrm{n}}(x')B_\mathrm{n}(x)\} \quad (3)$$

$$\dot{\mathbf{H}}_\mathrm{s} = -\frac{1}{2\pi} \int_S \int_{S'} \mathrm{d}^2x \mathrm{d}^2x' \hat{\mathbf{n}} \cdot \frac{x - x'}{|\,x - x'\,|^2}$$
$$\times \{[\mathbf{V}_{\perp\mathrm{t}}(x) - \mathbf{V}_{\perp\mathrm{t}}(x')]B_\mathrm{n}(x)B_\mathrm{n}(x')\} \quad (4)$$

Because the photospheric magnetic fluxes mainly consist of isolate photospheric flux concentrations, the shear term is primarily due to horizontal motions of the flux elements and the contribution from weak flux outside flux concentrations can be neglected. The shear term can thus be further decomposed in to a writhe term ($\dot{\mathbf{H}}_\mathrm{writhe}$) and a spin term ($\dot{\mathbf{H}}_\mathrm{spin}$), which refer to the contributions from the relative proper motions of photospheric flux elements about one another and from internal spinning motions within each element, respectively. Using $S_i$ to denote the region covering the $i$-th flux elements, the shear term can be expressed as[15,16]

$$\dot{\mathbf{H}}_\mathrm{shear} = \sum_i \dot{\mathbf{H}}_\mathrm{spin}^i + \sum_i \sum_{j \neq i} \dot{\mathbf{H}}_\mathrm{writhe}^{ij} \quad (5)$$

where

$$\dot{\mathbf{H}}_\mathrm{spin}^i = -\frac{1}{2\pi} \int_{S_i} \int_{S_i'} \mathrm{d}^2x \mathrm{d}^2x' \hat{\mathbf{n}} \cdot \frac{x - x'}{|\,x - x'\,|^2}$$
$$\times \{[\mathbf{V}_{\perp\mathrm{t}}(x) - \mathbf{V}_{\perp\mathrm{t}}(x')]B_\mathrm{n}(x)B_\mathrm{n}(x')\}, \quad (6)$$

and

$$\dot{\mathbf{H}}_\mathrm{writhe}^{ij} = -\frac{1}{2\pi} \int_{S_i} \int_{S_j} \mathrm{d}^2x \mathrm{d}^2x' \hat{\mathbf{n}} \cdot \frac{x - x'}{|\,x - x'\,|^2}$$
$$\times \{[\mathbf{V}_{\perp\mathrm{t}}(x) - \mathbf{V}_{\perp\mathrm{t}}(x')]B_\mathrm{n}(x)B_\mathrm{n}(x')\}. \quad (7)$$

Each spin term can be rewritten as

$$\dot{\mathbf{H}}_\mathrm{spin}^i \equiv -\frac{\Phi_i^2 \bar{\omega}_i}{2\pi}, \quad (8)$$

where

$$\bar{\omega}_i \equiv -\frac{2\pi \dot{\mathbf{H}}_\mathrm{spin}^i}{\Phi_i^2} = \frac{1}{\Phi_i^2} \int_{S_i} \int_{S_i'} \mathrm{d}^2x \mathrm{d}^2x' \hat{\mathbf{n}} \cdot \frac{x - x'}{|\,x - x'\,|^2}$$
$$\times \{[\mathbf{V}_{\perp\mathrm{t}}(x) - \mathbf{V}_{\perp\mathrm{t}}(x')]B_\mathrm{n}(x)B_\mathrm{n}(x')\}. \quad (9)$$

Here, $\bar{\omega}_i$ denotes a mean angular rotation rate weighted by the magnetic flux density in region $S_i$. Moreover, an alternative estimate of the mean angular rotation rate within the region $S_i$ can be directly obtained from the tangential velocity $\mathbf{V}_{\perp\mathrm{t}}$ as follows:

$$\bar{\omega}_i = \frac{\int_{S_i} \mathrm{d}^2x[(x - x_0) \times (V_{\perp\mathrm{t}}(x))]}{\int_{S_i} \mathrm{d}^2x}, \quad (10)$$

where $x_0$ is the centroid of region $S_i$.

**The changes in the Lorentz force.** Because the magnetic field distribution on the photosphere is not force-free, the Lorentz force is presumably balanced by other forces such as gas-pressure gradients and gravity when the atmosphere is approximately static. If the vector magnetic field significantly changes on a time-scale of a few minutes, such as in the course of a major flare, the abrupt change of the magnetic field could result in a change in the Lorentz force $\delta\mathbf{F}$ which could then produce an imbalance in the photosphere. The Lorentz force impulse thus applies to the photosphere until a new equilibrium is reached. The horizontal and vertical components of $\delta\mathbf{F}$ are expressed as[31]

$$\delta\mathbf{F}_\mathrm{h} = \frac{1}{4\pi} \int_S \mathrm{d}^2x \delta[B_\mathrm{r}(x)\mathbf{B}_\mathrm{h}(x)]$$
$$\delta F_\mathrm{r} = \frac{1}{8\pi} \int_S \mathrm{d}^2x[\delta B_\mathrm{r}^2(x) - \delta B_\mathrm{h}^2(x)] \quad (11)$$

When the horizontal Lorentz force applies to region $S_i$, the torque about an axis perpendicular to the surface through point $x_0$ can be expressed as

$$\delta\tau = \frac{1}{4\pi} \int_S \mathrm{d}^2x\{(x - x_0) \times \delta[B_\mathrm{r}(x)\mathbf{B}_\mathrm{h}(x)]\}, \quad (12)$$

where $x_0$ is taken as the centroid of region $S_i$.

**A step function fitting to the time variation of the magnetic field.** To characterize the impulsive changes of the measured parameters, we fit a step function[6]

of the following form[6] to all the time variation of the mean magnetic field of the sunspot:

$$B(t) = a + b\left\{1 + \frac{2}{\pi}\arctan[c(t - t_0)]\right\} \qquad (13)$$

where a, b, c and $t_0$ are the free parameters of the fit. The quantity $\pi c^{-1}$ is a measure of the period of time $\delta t$ over which the field change occurs. The amplitude of the step, $(4b)/(\pi)$, is a measure of the change in the magnetic field $\delta B$ within $\delta t$, and then the ratio of the amplitude of magnetic field change $\delta B$ to the amplitude of the magnetic field B is $(4b)/(\pi a)$. Similarly, we use the step function to fit the time variation of the mean photospheric $\alpha$ as well as the mean $\alpha$, the mean length and the mean twist of the NLFFF extrapolated field lines. The standard deviation of best fit parameters are used to estimate the uncertainties of the changes.

**Magnetic field modelling.** The magnetic field **B** in the corona is extrapolated with the help of a nonlinear force-free extrapolation code based on the optimization method[34–37]. The vector data at the photosphere is used as boundary condition for the extrapolation. The extrapolation calculation is performed within a box of $280 \times 264 \times 264$ uniform grid points, which corresponds to about $202 \times 190 \times 190\,\text{Mm}^3$.

The twist of a magnetic field line is expressed by $T_n = \frac{1}{4\pi}\int_0^L \alpha dl$ (ref. 64), where the scalar $\alpha = \mathbf{J}\cdot\mathbf{B}/|\mathbf{B}|^2$ is constant along a field line. However, since the values of $\alpha$ derived numerically are not completely invariant along a field lines of **B**, we adopt the average of $\bar{\alpha}$; along a field line,

$$\bar{\alpha} = \frac{1}{L}\int_0^L \alpha dl, \qquad (14)$$

and then

$$T_n = \frac{1}{4\pi}\bar{\alpha}L \qquad (15)$$

**Software availability.** DAVE-DAVE4VM flow tracking codes can be obtained from http://ccmc.gsfc.nasa.gov/lwsrepository/index.php; the feature tracking algorithm YAFTA can be downloaded from http://solarmuri.ssl.berkeley.edu/welsch/public/software/YAFTA/; the fit ellipse.pro procedure is available at https://www.idlcoyote.com/programs.

**Data availability.** All the data that are used in the current study are publicly available: The SDO/HMI vector magnetograms and SDO/AIA images that support the findings of this study are available on the Joint Science Operations Center (JSOC) website http://jsoc.stanford.edu; the GOES X-ray flux data is available from http://www.ngdc.noaa.gov/stp/satellite/goes/index.html

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

## Acknowledgements

The NASA/SDO data used here are courtesy of the HMI and AIA science teams. This work uses the DAVE/DAVE4VM codes written and developed by the Naval Research Laboratory and the NLFFF extrapolation code written by Dr Wiegelmann. Y.B. is grateful to Dan Yang for helpful discussions. This work is supported by the Natural Science Foundation of China under grants 11403098, 11173058, 11473065, 11273056, 11503081, 11503082 and 11633008.

## Author contributions

Y.B. analysed the data and wrote the text. Y.J. led the discussion. J.Y., J.H., H.L., B.Y. and Z.X. helped to improve the manuscript.

## Additional information

**Competing financial interests:** The authors declare no competing financial interests.

