## [Peer Review File · Nature Communications]

Reviewer #1 (Remarks to the Author):

This paper describes sudden photospheric magnetic field changes during a major solar flare, focusing on the relationship between proper sunspot rotational motion changes seen in HMI continuum intensity images and magnetic helicity and flow parameters derived from HMI vector magnetogram time series and the DAVE4VM optical flow code. The specific result that the sunspot rotation briefly changed direction during this flare is novel and merits publication. The related modeling result that the typical active region loop length/twist decreased/increased as a result of the flare is also interesting. The results are therefore worth publishing. The main changes that I would suggest are to add discussion of some closely-related work of the recent past, and to revisit the calculation of the orientation angle of the southern part of the sunspot of interest.

Relevant past work:

The authors may not be aware of some studies of sunspot magnetic twist changes, and related closed loop and flux rope changes, published in the recent past. Similar to Figure 4 in this paper, numerous studies based on nonlinear force-free modeling have produced evidence of collapsing twisted flux ropes during major flares (e.g., Sun et al. 2012; Liu et al. 2012, 2014; Jing et al. 2012).

Similar to the sunspot observations, Ruan et al. (2014) reported that circular sunspot motions may have energized the 6 September 2011 X2.1 flare in NOAA 11283, and Liu et al. (2014) concluded from nonlinear force-free modeling that a tilted magnetic flux rope, rooted in the rotating sunspot, crossed the main neutral line and collapsed during this X2.1 flare and the X1.8 flare that occurred the following day. Petrie (2013) found for the 15 February 2011 X2.2 flare in NOAA 11158 that abrupt untwisting forces occurred in two important sunspots located at opposite ends of the main neutral line, and Wang et al. (2014) found that the shear flows at the neutral line and circular motions at the two neighboring sunspots underwent sudden changes during this flare, consistent with the Lorentz force changes calculated from the vector magnetograms. (Petrie 2016 compared these two flares and concluded that they conform to different theoretical models, even though their photospheric behavior appeared very similar.)

The increase in magnetic twist found in the present paper seems to be a novelty though it resembles changes found near major photospheric neutral lines, where field vectors tend to become stronger, more horizontal, and more sheared shear there) during neutral-line flares, even while the other components tend to become more relaxed (Petrie 2012). The results in the present paper have strong similarities to these past results except that, besides the novel reversal of sunspot rotation direction, the penumbra became more twisted during this flare, more like the more-sheared neutral lines discussed above than the un-twisting sunspot fields. Some brief discussion of selected past results would help us to understand the results better and, I hope, broaden the appeal of the paper as well as emphasize its novelty. There may be other references that the authors care to add.

Southern sunspot ellipse calculation and interpretation:

Clarification is needed regarding the characterization of the nearly-ellipse-shaped southern part of the sunspot that is the focus of the study. Is the south part's northern boundary determined by the light bridge or another feature? Could the measured change in ellipse orientation have been caused by movement or change in the light bridge structure, or strengthening or weakening of penumbral structure, a phenomenon that has been reported many times in the past, rather than global rotation of this part of the sunspot? Given the collapsing loop structure reported in the paper, some 'permanent' strengthening of penumbral structure could have occurred. This could alter the orientation of a fitted ellipse in a stepwise fashion without a global rotation of the

structure having occurred.

Furthermore, the estimated effect of the Lorentz torque change on the orientation of this part of the sunspot seems to rely on an estimate of the moment of inertia of the structure, the details of which are not given. This would involve an estimate of density and volume of the structure affected, and would have to take into account the likely strong magnetic connections to the much denser solar interior.

For these reasons the observation of abrupt orientation change and the connection to the Lorentz torque change, though tantalizing, may not be as simple as represented in the text. In addition, in Figure 2 the ellipse orientation profile (a, top) resembles the DAVE4VM mean angular speed profile (a, bottom), and not its integral. Can the authors clear up these issues or discuss the affected results in more cautious terms?

Some more minor suggestions follow.

Though the soft X-ray flux may be plotted in Figure 3, the paper does not identify the flare's GOES class or peak (1-8 Å) flux.

p2 col 1 para 3: on within 24 minutes -> within 24 minutes

p2 col 1 para 4: ...and then could produce upward and vertical torque - I didn't understand how "vertical torque" could be inferred from Figure 1d.

which is consistence with -> which is consistent with

p2 col 2 last para: exerting to the sunspot -> exerted on the sunspot

In Figure 3 there is a wide plot of what looks like soft X-ray flux (without vertical axis label and units, and the caption says that panel a shows helicity density maps), followed by a row of two pairs of spatial plots, then two wide plots of helicity flux against time. Would it make aesthetic and ergonomic sense to put the row of spatial plots at the top and then have the temporal plots stacked together? Of course this is just a suggestion.

p8 III: mothedcs -> methods (also in title of C)

p9 F: The nonlinear force-free extrapolation code is said to be based on the optimization method of Wiegelmann et al. Have details of the new code been published elsewhere? If so, a reference would be helpful.

Reviewer #2 (Remarks to the Author):

The authors present a case where the rotation of a sunspot is found to reverse after a flare. The authors interpret this as being due to the change in the magnetic field after a flare exerting a Lorentz force across the photosphere which then drives counterrotation of the spot (to a depth of 250km).

The authors have used several analysis methods to show that change in rotation is real, and this part is nicely done. However the interpretation that the change in rotation is due to the Lorentz force rests on the assumption that Lorentz force needs only be applied to about 250km in the photosphere -- which is problematic.

It is true that the increasing density beneath the photosphere quickly decreases the Alfvén velocity, however as they note the scale height where they are interested is also about 250km,

which means the decrease in the Alfvén velocity is modest. In all reasonable models of the sunspots, the field is sufficiently strong that the Alfvén velocity is greater than the sound speed in about the top 500 Km, which means that the Alfvén transit time will be about $250\text{km}/10\text{km/s}=25$ sec. So it is difficult to see how the proposed change in angular momentum in the top 250Km can remain decoupled from the underlying magnetic flux for longer than this. I regard this as a major problem with the proposed interpretation.

Secondly the authors dismiss the possibility that the change is being driven from below on the basis that the Alfvén transit time is too long 'to be compatible'. I don't see the basis for this -- the main timescale of interest is about 20 minutes, why is this too short?

I also wish to point out that Figure 2 is mislabeled -- the labels b & c need to be exchanged.

Reviewer #3 (Remarks to the Author):

The paper reports excellent observations and analysis of a remarkable event that appears to establish, with great significance, the existence of a tangible back-reaction on the solar photosphere from a coronal magnetic-field restructuring. As the best example of this physically very plausible phenomenon, it warrants publication after the small details pointed out here are corrected. I will not need to review the paper a second time.

Minor corrections:

Please call the flare SOL2014-09-10 and note its heliographic location and GOES class.

The assumption about the mass of material needs to be stated in the paper as well as in the Methods section.

Typo "serious" should be "series".

Typo "motheds" should be "methods".

Reviewer #4 (Remarks to the Author):

SUMMARY OF THE KEY RESULT: The authors report changes in the rotation of a sunspot around the time of a flare, and assert this reversal is driven by a change in Lorentz force due to the flare.

ORIGINALITY AND INTEREST: This manuscript makes, to my knowledge, the first report of a reversal of rotation associated with a flare. This is probably of interest to a subset of the solar physics community.

That said, the significance of these observations arises primarily from their novelty: beyond being the first such observation, I do not think most solar researchers will be very surprised by it, nor do I think it will lead to any re-evaluation of theories regarding solar flares (i.e., these observations are consistent with current understanding of solar magnetic field dynamics and flares).

Many previous studies (including those cited in the article --- by Anwar et al. and Wang et al. in the 1990s, and more recently by Sudol & Harvey and others) have found flare-associated changes

with sunspots and solar magnetic fields. The observations reported here are a type of flare-related magnetic field change not reported before, but the implications of these observations are not, to my mind, substantially different from those discussed in previous reports. The observations discussed in the cited paper by Wang et al. (2014) of flare-related changes in rotation are quite similar (but do not show a reversal of rotation).

Reversals in the direction of rotation of sunspots that were not associated with flares have also been reported before, for instance: <http://adsabs.harvard.edu/abs/2004AAS...204.3716N>
<http://adsabs.harvard.edu/abs/2011ApJ...729...95G> These reports should be cited in the paper.

If the authors believe there are more significant implications from their observations, they should highlight these more strongly.

DATA & METHODOLOGY:

VALIDITY OF APPROACH, QUALITY OF DATA, AND QUALITY OF PRESENTATION:

A terse summary of the paper's approach is: the authors saw a reversal spot rotation; this happened at the same time as a flare; the authors assert that this correlation implies causation, i.e., that the flare caused the change in rotation. While this interpretation is very plausible to me (and in fact I suspect it is correct), the causal link has not been proven. (Nor could it be, without additional information about the magnetic field's structure and evolution in both the solar interior and atmosphere that cannot be obtained with present observational capabilities.) So the authors should note that while a causal link is physically plausible, such a link is an unproven (but clearly favored) hypothesis. The authors should clearly describe any alternative hypothesis of which they are aware. Near the top of the right column of p.2, the text states flare-associated changes to the photospheric field are "more likely" --- but more likely than what? Perhaps they mean something like, "Rapid changes in the photospheric field and motions at the photosphere during the flare are more likely driven by the flare than by convective flows, since the photospheric convective turnover time at this length scale is much longer than the timescale of the changes." Note that I substituted "convective timescale" for "Alfvénic timescale" here: if the photospheric field changes are driven by the coronal evolution during the field, these changes must have been mediated by Alfvén waves that propagated down into and then within the photosphere.

The authors' reasoning behind the following statement is unclear: "we obtain the change of the torque during the flare to be..." (p.2, bottom left) and "we can deduce that the torque impulse could cause the observed angular velocity of $20^\circ/\text{hr}$ " (below equation 11). Torque on a point mass is $(r \times F)$, and torque on an extended object requires an integration that includes a factor of r for the moment arm of the force. But the authors only give expressions for F (equation 11). What is the length scale used in computing the torque? In particular, I believe the authors need something like: $\int dx dy (r \times \Delta F)$. This integration should be discussed in the text, assuming it was performed. (The authors could mention that the already cited paper by Wiegmann (2012), on its p. 41, has an expression containing components of the static torque that can be adapted to give the first-order torque arising from changes in B .) How was the moment of inertia estimated? The cited paper by Wang et al. (2014) discusses the choice of rotation axis and estimate of the moment of inertia in detail (on its third page). The angular acceleration mentioned on p.2, $1.4 \text{ degree}/\text{hour}/\text{second}$, multiplied by the 30 seconds mentioned at the end of the same paragraph (and on p.9) would yield a rotation rate of about $40 \text{ deg}/\text{hr}$, which is twice the fitted $20 \text{ deg}/\text{hr}$. How does this discrepancy arise? Is the $1.4 \text{ deg}/\text{hr}/\text{sec}$ figure a peak value, such that the average angular acceleration is half this? I note that based upon the bottom plot of FIG. 1 panel c, the Lorentz torque appears to act for much longer than 30 sec -- perhaps for as long as 300 sec! The authors could address this concern by including a bit more info about their procedures in III. D. on p. 9.

I have no concerns about the quality of the data used: the authors used vector magnetic field and atmospheric imaging measurements from the SDO satellite, which have been employed by many others in studies of solar phenomena. (If magnetogram data from Hindoe/SOT/SP were available, the uncertainties in the magnetic field measurements would likely be reduced, but SP data have limited coverage and very low cadence relative to flare dynamics).

LACK OF USE OF STATISTICS AND TREATMENT OF UNCERTAINTIES: Plots that might have included uncertainties or estimates of variance (e.g., FIG. 2, FIG. 3, FIG. 4) do not include them. Rotation rates (given in degrees / hr) were also presented without uncertainty estimates. This omission is particularly serious in the reported 20-degree/hr jump associated with the flare. How susceptible is this result to different choices of start/stop times, or different parameters used to fit the ellipse? FIG. 2. panel (a) shows the fitted slope, but what are the uncertainties of this fit?

CONCLUSIONS: The authors' argue that the flare caused the change in sunspot rotation. This is very plausible, but cannot be conclusively shown with available observations: previous observations of changes in sunspot rotation that were not associated with flares imply that forces arising from within the solar interior could have altered the the spot's rotation, independent of the flare. Accordingly, the authors should rephrase passages of the text that suggest the link has been proven to note the possibility (however remote) that some other causal mechanism is responsible.

To summarize the most significant changes that must be made:

(a) Include uncertainties (for data, fits to data, or quantities derived from data -- like helicity flux), or variances (for things like the average lengths of field lines, or their average twists), in all measured or inferred quantities.

(b) Mention that the observed temporal correlation between the flare and change in spot rotation does not prove causation. The authors certainly can state that they favor the flare-as-cause hypothesis. (I do.)

(c) Include more description of how the magnetic torque on the sunspot was estimated.

Beyond these specific issues, more minor scientific issues are:

(d) At the end of the first paragraph on p.1, I am surprised by the authors' outward-biased view: they mention "the role the spinning motion can play in injecting helicity into the corona during a flare." Elsewhere in the paper, however, the authors argue that the changes in photospheric field and forces/torques are driven by the flare -- which originates in the corona. Since the flare is very short compared to timescales of variation of the photospheric field outside of the flaring time interval, I expect spinning motion driven from the interior plays almost no role in helicity transfer during the flare. Because helicity can cross the photosphere in either direction, the text should reflect this: for instance, "the role the spinning motion can play in transferring helicity between the interior and corona (in either direction) during a flare."

(e) p.1, at the first discussion of the flare times (upper-right column), please mention the flare's GOES class. From FIG. 3, it looks > X1.0.

(f) Equation (5) in the appendix is incorrect: there should be two integrals, both over the domain S_i (one primed and one unprimed). As written, the units are wrong -- the numerator needs a factor of length^2 to be a helicity. As a related issue, the domains of the integrations in equation (6) should clearly be labeled as distinct --- e.g., S_j and $S_{\{i \neq j\}}$. As with equation (5), equation (8) needs two integrals, both over S_i (one primed and one unprimed).

(g) Equation (9) is dimensionally flawed: the integral should be normalized by the area integrated in order for the angular velocity to have units of radians per second.

(h) Also, while equation (9) (properly normalized) might yield values close to equation (8), in general the angular velocities computed in these different ways should not be equal: the integrand in equation (8) is weighted by the magnetic flux density, but in (9) it is not. Accordingly, the introductory sentence should be rewritten: "Moreover, an alternative estimate of the mean angular rotation rate within..." (Note also that this sentence should conclude with "as follows:" instead of "as follow")

(i) Near the end of p.1, why is I the difference between the rotation estimates from DAVE4VM (c. 1-3 deg/hr) and ellipse fitting (c. 20 deg/hr) so big? One possibility is that a slight submergence of tilted field (due to inward contraction of the magnetic field as a result of the flare) can lead to a large "apparent footpoint motion" like that discussed by Demoulin & Berger (2003), <http://adsabs.harvard.edu/abs/2003SoPh..215..203D> -- see their Fig. 1, but with submergence instead of emergence. I believe this manuscript's observations might support this apparent footpoint model.

(j) Toward the bottom of the left column of p.2 the text states: "could produce upward and vertical torque". Do the authors mean a torque about the vertical axis, in the upward direction? I believe this would yield a counter-clockwise rotation, but the text discusses clockwise rotation. A clockwise rotation would arise from downward torque about the vertical axis.

(k) In the right col. of p.2, the authors mention field lines in the NLFFF model shortening. I think it is appropriate to repeat the citation of Hudson's (2000) paper on coronal implosion at this point.

(l) p.2: "impulsively, the change"  "impulsively, due to the low Alfvén speed in the interior, the change"

(m) FIG. 1, panels (c) and (d): what is the scale of the V_t , B_t , and dB_t at upper left in each image? This could be stated in the caption, to avoid redoing the figures; but these figures would be more understandable if the scale for these vectors were in the figure itself.

(n) I don't understand the following statement: "Since no significant translational motions was [sic] detected neither in the centroid of P or in the centroid of O in the course of the flare, the impulsive change in \dot{H}_{PO} writhe should also be generated by the abrupt rotation of the sunspot." If the centers of flux of P and O don't move, then they don't wind about each other. So how does this writhe helicity come about?

(o) For equation (12) in III. E., the first place I am aware of seeing the functional form used was the cited paper by Sudol & Harvey (2005) [6]. I think it should be cited again here.

REFERENCES: While I infer the references were meant to be numbered in the order cited in the paper, they were not always --- see, e.g., [13] then [12] in the manuscript's first paragraph. Citation [18] has no title in the references.

A reference to the Sudol & Harvey paper (reference [6] in the manuscript) should be added in the paper's first paragraph, where citations [2,3,9] are already present.

In addition to the issues mentioned above, many minor instances of poor grammar, diction, and misspellings must be corrected. (I recommend that the authors run their manuscripts through a spell-checking routine prior to submitting them.) These are enumerated below.

Abstract: "back actions"  "back reactions"

Abstract: "It provides solid evidence..." (unclear referent of "It")  "These observations provide solid evidence..."

Abstract: "a change of Lorentz force exerting on the sunspot"  "a change of Lorentz force exerted on the sunspot"

Abstract: "It support the view"  "These observations support the view"

Abstract: "that the injection of the impulsive helicity flux" is a dynamic process respond to"
 either:

"that helicity is impulsively injected as a dynamic response to" or "that the inferred impulsive helicity flux is a dynamic response to"

Here, "dynamic response to" could be replaced with "dynamic process, in response to", but the latter phrasing uses more words without conveying more information.

p.1: "In particular, the abrupt"  "In particular, abrupt"

p.1: "motion of the magnetic flux"  omit "the"

p.1: "motion of magnetic flux would lead to"  "motion of magnetic flux could lead to" (Not all motions inject helicity. For instance, uniform translation does not.)

p.1: "It was reported in the course of some major flares that "  "It was reported that, in the course of some major flares, "

p.1: "the magnetic helicity was impulsively"  omit "the"

p.1: "across the photosphere and"  insert comma before "and"

p.1: "and the helicity flux tends"  "and that the helicity flux tended"

p.1: "some literatures report"  change "literatures" to "researchers" or "papers"

p.1: "reported the sudden shear-relaxing"  "reported sudden, shear-relaxing"

p.1: "course of the flare and the motions"  "course of flares. These motions..."

p.1: "that the shear-relaxing motion is possible to play"  "that such shear-relaxing motions possibly play"

p.1: "within an isolate"  "within an isolated"

p.1: "is another kind of the photospheric tangential motions." 
"is another kind of photospheric tangential motion."

p.1: "which can be regard as"  "which can be regarded as"

p.1: "a process that transfer magnetic" 
"a process that transfers magnetic"

p.1: "Here, we determined the suddenly reversal" 
"Here, we characterized the sudden reversal"

p.1: "45 seconds cadence"  "45-second cadence"

p.1: "12 minutes cadence vector magnetogram" 
"12 minute cadence vector magnetograms"

p.1: "which is a mature"  "is" to "was"

p.1: "The intensity image (Fig. 1b)"  insert "photospheric" after "The"

p.1: Fig. 1b is discussed before Fig. 1a. Since 1a. shows emission during the flare, I suggest re-ordering the image panels to match their presentation in the text, or re-ordering the text to mention the flare first.

p.1: "into the southern and northern part." 
"into southern and northern parts."

p.2: "It suggests that no..."  "This suggests that no..."

p.2: "It indicates that the impulsive..." 
"This suggests that the impulsive..."

p.2: "we perform B_t in polar"  "we represent B_t in polar" or "we transform B_t into polar"

p.2: "r = 0 locating in"  "r = 0 located in"

p.2: "represented in the radial"  "represented as a"

p.2: "and azimuthal"  "and an azimuthal"

p.2: "changes in B_t was"  change "was" to "were"

p.2: "change of horizontal Lorentz force change acting"  "change in horizontal Lorentz force acting" (omit 2nd change)

p.2: "could produce upward and vertical torque" 

p.2: "which is consistence"  "which is consistent"

p.2: "Using the centroid of the sunspot as the pivot"  "With the centroid of the sunspot as the axis"

p.2: "flare amounts to be"  "flare to be"

p.2: "order higher"  "order of magnitude"

p.2: "It indicates that the sunspot could accelerated to..."  "This indicates that the sunspot could accelerate to..." (or could be accelerated to)

p.2: "lasted for 30 second."  "lasted for 30 seconds."

p.2: "on the photosphere at the flare time"  either "at" or "of" or "in" the photosphere, and "during the flare", since the flare lasts for a finite interval of time

p.2: "a impulsive"  "an impulsive"

p.2: "that are originated from P evolve to have higher" 
"that originate from P evolve toward higher"

p.2: "fields is possible to be resulted"  "fields possibly resulted"

p.2: "it gives"  "the models suggest"

p.2: "field lines even if"  "field lines, even if"

p.2: "twist of the field line is"  "twist of the field is"

p.2: "The rotation of the sunspot"  "Outside of flares, the
rotation of sunspots"

p.2: "photosphere and then results"  "photosphere, which results"

p.2: "force exerting to the sunspot."  "force exerted on the sunspot."

FIG. 1a. caption: "AIA image at"  "AIA images at"

FIG. 1a. caption: "chromopheres"  "chromosphere"

FIG. 1a. caption: "the flaring loop"  "flaring loops"

FIG. 1b. caption: "intensity image"  "intensity images"

FIG. 1b. caption: "left panels"  "left panel"

FIG. 1b. caption: "close-up view"  "close-up views"

FIG. 1b. caption: "refer to the contour with the contour level of 13000"  "refer to the 13000-
data-number contour level." (I assume the units of 13000 are DN.)

FIG. 1b. caption: "which outline"  "which outlines"

FIG. 1b. caption: "by best fitting"  "by fitting"

FIG. 1c. caption: "The image is the"  "The background grayscale
image shows the"

FIG. 1c. caption: "vertical field with"  "vertical field, with"

FIG. 1c. caption: "On the right two panels, the close-up view"  "In the right two panels, close-
up views"

FIG. 1d. caption: "The image is the vertical field same as in c, on which the superimposed blue
arrows refer to the tangential field vectors, while the red arrows on the middle panels refer to
changes of the tangential field and the red arrows on the right panel refer to the changes of the
tangential Lorentz force."

"As in c, the background grayscale shows the vertical field. In the middle panel, the superimposed
blue arrows refer to the tangential field vectors, while the red arrows on the middle panels refer to

changes of the tangential field. In the right panel, the red arrows refer to the changes in the tangential Lorentz force."

FIG. 1d. caption: "On each panels, the blue curve outlines a region labeled by P, which matches a flux concentration on the total field images and is identified by a named Yet Another Feature Tracking Algorithm (YAFTA)."

"The blue curve in each panel outlines the region that we label P, which corresponds to a flux concentration (in images of the total magnetic field) identified with the Yet Another Feature Tracking Algorithm (YAFTA) package." [So this feature was identified in images of $|B|$, not B_z , correct?]

FIG. 2a. caption: "kinematic parameter"  "kinematic parameters"

FIG. 2a. caption: "is best-fit"  "best fits"

FIG. 2a. caption: "dash line refers"  "dashed lines refer"

FIG. 2a. caption: "The positive (resp. negative)"  "Positive (resp. negative)"

FIG. 2a. caption: "the counter-clockwise (resp. clockwise)"  "counter-clockwise (resp. clockwise)"

FIG. 2b and 2c captions: these were interchanged; please switch them.

FIG. 2b and 2c captions: capitalize start of first sentence, "the"  "The"

FIG. 2b. caption: "exerting to"  "exerted on"

Fig. 3. caption: The labels for each panel are incorrect. Panel (a), the GOES light curve, is not described at all.

Fig. 3a caption: The text after (a) in the current caption describes (b). "14-hours averaged"  "14-hour averaged"

Fig. 3b caption: This caption describes (c). Capitalize first word, "the".

Fig. 3c caption: This caption describes (d). Capitalize first word, "the".

Fig. 3c caption: "over the region as the FOV of panel (a)"  "over the same FOV as panel (b)"

Fig. 3d caption: This refers to the panel labeled (e). Capitalize the first word, "the." Later, "the blue line refer to"  "The blue line refers to"

Fig. 3e caption: I think this refers to a panel that does not exist.

p. 8, III. (section title): "MOTHEDS"  "Methods"

p. 8, III.A., "HMI takes the full-disk"  omit "the"

p. 8, III.A., "[32] and the remaining"  "[32]. The remaining"

p. 8, III.A., "the mean"  "a mean"

p. 8, III.B., "is ellipse"  "is elliptical"

p. 8, III.B., "of the ellipse that best-fit it"  "of the best-fit ellipse"

p. 8, III.B., "it starts"  "it started"

p. 8, III.B., "several hours after"  "until several hours after"

p. 8, III.B., "It indicates"  "This indicates"

p. 8, III.C. (subsection title): "Motheds of helcity" (two misspellings)  "Methods for helicity fluxes"

p. 8, III.C., "donates the vector"  "denotes the vector"

p. 8, III.C., "components of velocity V_{\perp} , the velocity"  "components of V_{\perp} , the velocity"

p. 8, III.C., "to the magnetic field lines."  "to the magnetic field."

p. 8, III.C., "from the the twisted"  omit "the the", so "from twisted"

p. 8, III.C., "by the tangential"  "by tangential"

p. 8, III.C., "fluxes are mainly contributed by the isolate Photospheric"  "fluxes mainly consist of isolated photospheric"

p. 8, III.C., "contributed by the horizontal motions of the flux elements"  "due to horizontal motions of flux elements"

p. 8, III.C., "from the weak flux outside of the flux concentrations"  "from weak fields outside flux concentrations"

p. 8, III.C., "can be negligible."  "can be neglected."

p. 8, III.C., interchanged reference:

"the spin term (H_{sp}) and writhe term (H_{wh}), which refers to the contribution from relative proper motions of photospheric flux elements about one another and from internal spinning motions within each element, respectively."

 switch H_{sp} and H_{wh} to match descriptions,

"the writhe term (H_{wh}) and spin term (H_{sp}), which refers to the contribution from relative proper motions of photospheric flux elements about one another and from internal spinning motions within each element, respectively."

p. 9, III.C., "we define the region as "P", which is tracked by a named Yet Another Feature Tracking Algorithm (YAFTA) [14] and encloses a flux concentration covering the rotational part of the"

"we used the Yet Another Feature Tracking Algorithm (YAFTA) package [14] to define and track the region labelled "P". This region encloses a flux concentration that comprises the rotational part of the"

p. 9, III.C., "sunspot, while the region outside the region 'P' is" 
"sunspot. The region outside 'P' is"

p. 9, III.C., "the equation (4)"  omit "the"

p. 9, III.C., "motions was detected neither in the centroid of P or in the centroid of O" 
"motions were detected in the centroids of either of P or O"

p. 9, III.C., "Indicated"  "As indicated"

p. 9, III.C., "but the reversal"  omit "but"

p. 9, III.D., "the Lorentz force are"  "are" to "is"

p. 9, III.D., "pressure gradient"  "pressure gradients"

p. 9, III.D., "and then produce"  "which could then produce"

p. 9, III.D., "We makes the simple assumption"  "We make the simplifying assumption"

p. 9, III.D., "time delta t, over which"  omit comma

p. 10, III.F., "is used to be the"  "is used as"

p. 10, III.F., "twist of magnetic"  "twist of a magnetic"

Dear Reviewer:

Thank you for your comments concerning our manuscript. Those comments are all valuable and very helpful for revising and improving our paper, as well as the important guiding significance to our researches. We have studied comments carefully and have made correction, which we hope meet with approval.

In the file of "changes_are_marked.pdf", the portions revised by authors are marked in blue letters and the portions corrected by the reviewer are marked in red letters. The main corrections in the paper and the responds to your comments are as following

Reviewers' comments:

Reviewer #1 (Remarks to the Author):

This paper describes sudden photospheric magnetic field changes during a major solar flare, focusing on the relationship between proper sunspot rotational motion changes seen in HMI continuum intensity images and magnetic helicity and flow parameters derived from HMI vector magnetogram time series and the DAVE4VM optical flow code. The specific result that the sunspot rotation briefly changed direction during this flare is novel and merits publication. The related modeling result that the typical active region loop length/twist decreased/increased as a result of the flare is also interesting. The results are therefore worth publishing. The main changes that I would suggest are to add discussion of some closely-related work of the recent past, and to revisit the calculation of the orientation angle of the southern part of the sunspot of interest.

Relevant past work:

The authors may not be aware of some studies of sunspot magnetic twist changes, and related closed loop and flux rope changes, published in the recent past. Similar to Figure 4 in this paper, numerous studies based on nonlinear force-free modeling have produced evidence of collapsing twisted flux ropes during major flares (e.g., Sun et al. 2012; Liu et al. 2012, 2014; Jing et al. 2012).

RESPONSE: In this reversion, the references are numbered as [44-48] (see 4th paragraph in Discussion)

Similar to the sunspot observations, Ruan et al. (2014) reported that circular sunspot motions may have energized the 6 September 2011 X2.1 flare in NOAA 11283, and Liu et al. (2014) concluded from nonlinear force-free modeling that a tilted magnetic flux rope, rooted in the rotating sunspot, crossed the main neutral line and collapsed during this X2.1 flare and the X1.8 flare that occurred the following day. Petrie (2013) found for the 15 February 2011 X2.2 flare in NOAA 11158 that abrupt untwisting forces occurred in two important sunspots located at opposite ends of the main neutral line, and Wang et al. (2014) found that the shear flows at the neutral line and circular motions at the two neighboring sunspots underwent sudden changes during this flare, consistent with the Lorentz force changes calculated from the vector magnetograms. (Petrie 2016 compared these two flares and concluded that they conform to different theoretical models, even though their photospheric behavior appeared very similar.)

RESPONSE: In this reversion, In this reversion, the references are referred in Discussion

The increase in magnetic twist found in the present paper seems to be a novelty though it resembles changes

found near major photospheric neutral lines, where field vectors tend to become stronger, more horizontal, and more sheared shear there) during neutral-line flares, even while the other components tend to become more relaxed (Petrie 2012). The results in the present paper have strong similarities to these past results except that, besides the novel reversal of sunspot rotation direction, the penumbra became more twisted during this flare, more like the more-sheared neutral lines discussed above than the un-twisting sunspot fields. Some brief discussion of selected past results would help us to understand the results better and, I hope, broaden the appeal of the paper as well as emphasize its novelty. There may be other references that the authors care to add.

RESPONSE: As the review suggested, the results in the present paper show the penumbra became more twisted during this flare, like the more-sheared neutral lines discussed above than the un-twisting sunspot fields. We discuss this similarity as follows:

“It has been reported that abruptly decrease in azimuthal field occurred in two sunspots in NOAA 11158 during an X2.2 flare [32] and the corresponding Lorentz force changes were consistent with the stepwise increased rate of sunspot rotation [7]. The authors noted that the sunspots may connect the two ends of the flux rope associated with the flare, and then suggested that the decrease in horizontal field of the sunspots could be explained by that some of the twist component of the field was eliminated from the flux rope [33]. Different from their studied case, the sunspot presented here was swept by the flare ribbon. Therefore, the physical mechanism evolving in the sunspot during the flare should be similar to that evolving in the magnetic flux along the flaring PIL, at which a significant increase of transverse field and a relative small change in the vertical field is often found. Such change in photospheric field implies that there are Lorentz-force changes acted on the photosphere, which trends to relax the shear of the field near the PIL. On the other hand, the enhancement of transverse field near PIL is often accompanied by an increase of magnetic shear at the core flaring region [34].”(also see 2nd paragraph in Discussion)

We also note that coronal magnetic field originated from the reported sunspot shrinks with increasing twist density. Similarly, the past works also found the increasing shear along PIL. Here, we point out that the reversal of rotation in the sunspot as a process of magnetic twist transportation from the corona into the solar interior, supporting that the enhancement in the local twist density could occur in the corona impulsively during the flare.

Southern sunspot ellipse calculation and interpretation:

Clarification is needed regarding the characterization of the nearly-ellipse-shaped southern part of the sunspot that is the focus of the study. Is the south part's northern boundary determined by the light bridge or another feature? Could the measured change in ellipse orientation have been caused by movement or change in the light bridge structure, or strengthening or weakening of penumbral structure, a phenomenon that has been reported many times in the past, rather than global rotation of this part of the sunspot? Given the collapsing loop structure reported in the paper, some 'permanent' strengthening of penumbral structure could have occurred. This could alter the orientation of a fitted ellipse in a stepwise fashion without a global rotation of the structure having occurred.

RESPONSE:[1] Is the south part's northern boundary determined by the light bridge or another feature? we find that the 13000-data-number contour level on the intensity images approximately outline the umbra-penumbral boundary of the sunspot.(see)

[2] " After carefully examining the intensity images, we find that all of the contour lines at

contour levels of data-number varying between 13000 and 17000 could match well the umbra-penumbral boundary of the sunspot on the intensity images taken around the time of the flare. Supplementary Movie 1 shows the evolution of ellipses fitting the 10 different data-number contour lines. The selected data-numbers range from 13000 to 17000 and space apart from one another at regular interval of 500. As showed in Supplementary Movie 1, all of the fitting ellipses appear to undergo a sudden clockwise rotation during the flare although the details of the contour lines are different with each other. Therefore, it is possible that the measured change in ellipse orientation was caused by global rotation of this part of the sunspot instead of the movement or change in the umbral structure. In Fig. 3d, the points mark the orientations of the ellipses fitting different contour levels at each moment. It clearly shows that the data points taken around 17:36 UT are rather close to each other, indicating that the ellipses fitting different contour levels rotate in an identical way in this period. Thus, it seems that the changes in the orientations of the ellipses exactly mark the rotation of the sunspot around 17:36 UT. " (also see section D in Result)

Furthermore, the estimated effect of the Lorentz torque change on the orientation of this part of the sunspot seems to rely on an estimate of the moment of inertia of the structure, the details of which are not given. This would involve an estimate of density and volume of the structure affected, and would have to take into account the likely strong magnetic connections to the much denser solar interior.

RESPONSE: To estimate the moment of inertia of the sunspot, firstly, we estimate the period of the angular acceleration for the best-fit ellipses amount to about 2 minutes. "Accordingly, the torsional Alfvén pulse could propagate to a depth of about 0.5 Mm below the photosphere within 2 minutes, during which the sunspot is detected to undergo an angular acceleration as discussed above. Thus, the Lorentz torque could apply on the underlying magnetic flux with the depth of 0.5 Mm." The flux tube involved in the reverse rotation would then have the moment of inertia of order of $2 \times 10^{30} \text{ N m}^2$. (also see 2nd paragraph in section F in Result)

For these reasons the observation of abrupt orientation change and the connection to the Lorentz torque change, though tantalizing, may not be as simple as represented in the text. In addition, in Figure 2 the ellipse orientation profile (a, top) resembles the DAVE4VM mean angular speed profile (a, bottom), and not its integral. Can the authors clear up these issues or discuss the affected results in more cautious terms?

RESPONSE: in this revision, we present numerical differentiation to the 12-minutes-average orientations of the ellipse in Fig. 3b. It can be seen that the temporal profile of rotation rate of the ellipse is consistent with the DAVE4VM mean angular speed profile. (see Fig. 3a and 3b)

Some more minor suggestions follow.

Though the soft X-ray flux may be plotted in Figure 3, the paper does not identify the flare's GOES class or peak (1-8 Å) flux.

RESPONSE: The flare's class is X1.6, which can be seen in last paragraph on page 1 in this revision

p2 col 1 para 3: on within 24 minutes -> within 24 minutes

RESPONSE: It has been changed

p2 col 1 para 4: ...and then could produce upward and vertical torque - I didn't understand how "vertical torque"

could be inferred from Figure 1d.

RESPONSE: After carefully examining the change in Lorentz force, we find that the torque about the vertical axis is downward. We changed the sentence that "... upward and vertical torque ..." into "The force acted on the sunspot in the clockwise direction and then could produce a torque about the vertical axis in the downward direction."

which is consistency with -> which is consistent with

RESPONSE: It has been changed

p2 col 2 last para: exerting to the sunspot -> exerted on the sunspot

RESPONSE: It has been changed

In Figure 3 there is a wide plot of what looks like soft X-ray flux (without vertical axis label and units, and the caption says that panel a shows helicity density maps), followed by a row of two pairs of spatial plots, then two wide plots of helicity flux against time. Would it make aesthetic and ergonomic sense to put the row of spatial plots at the top and then have the temporal plots stacked together? Of course this is just a suggestion.

RESPONSE: In this reversion, we have deleted the row of spatial plots in this Figure and then temporal plots stacked together.

p8 III: methods -> methods (also in title of C)

RESPONSE: It has been changed

p9 F: The nonlinear force-free extrapolation code is said to be based on the optimization method of Wiegmann et al. Have details of the new code been published elsewhere? If so, a reference would be helpful.

RESPONSE: Details of the code have been published in the references[30-32]

Reviewer #2 (Remarks to the Author):

The authors present a case where the rotation of a sunspot is found to reverse after a flare. The authors interpret this as being due to the change in the magnetic field after a flare exerting a Lorentz force across the photosphere which then drives counterrotation of the spot (to a depth of 250km).

The authors have used several analysis methods to show that change in rotation is real, and this part is nicely done. However the interpretation that the change in rotation is due to the Lorentz force rests on the assumption that Lorentz force needs only be applied to about 250km in the photosphere -- which is problematic.

It is true that the increasing density beneath the photosphere quickly decreases the Alfvén velocity, however as they note the scale height where they are interested is also about 250km, which means the decrease in the Alfvén velocity is modest. In all reasonable models of the sunspots, the field is sufficiently strong that the Alfvén velocity is greater than the sound speed in about the top 500 Km, which means that the Alfvén transit time will be about $250\text{km}/10\text{km/s}=25$ sec. So it is difficult to see how the proposed change in angular momentum in the top 250Km can remain decoupled from the underlying magnetic flux for longer than this. I regard this as a major problem with the proposed interpretation.

RESPONSE: in this revision, we give an estimate for the length of the underlying magnetic flux coupled with the angular momentum. Taking advantage of the 45-second cadence intensity images, we note that the spin motion of the sunspot was accelerated within 96 ± 20 seconds, from 17:29 UT to 17:31 UT (see 3rd paragraph in section D in Result). We adopt the Model S to estimate the density below the photosphere, and then we obtain that the torsional Alfvén pulse could propagate to a depth of about 0.5 Mm below the photosphere in the period of 2 minutes (see 2nd paragraph in section F in Result).

Secondly the authors dismiss the possibility that the change is being driven from below on the basis that the Alfvén transit time is too long 'to be compatible'. I don't see the basis for this -- the main timescale of interest is about 20 minutes, why is this too short?

RESPONSE: the Alfvén transit velocity immediately below the photosphere could be close to 10 Km/s. The sentence that "Alfvén transit time is too long to be compatible" is wrong as pointed out by the reviewer. Here, we delete the sentence, and we suggest that "Rapid changes in the photospheric field and motions at the photosphere during the flare are more likely driven by the flare than by convective flows, since the photospheric convective turnover time at this length scale is much longer than the timescale of the changes." (see 1st paragraph in Discussion)

I also wish to point out that Figure 2 is mislabeled -- the labels b & c need to be exchanged.

RESPONSE: It has been changed

Reviewer #3 (Remarks to the Author):

The paper reports excellent observations and analysis of a remarkable event that appears to establish, with great significance, the existence of a tangible back-reaction on the solar photosphere from a coronal magnetic-field restructuring. As the best example of this physically very plausible phenomenon, it warrants publication after the small details pointed out here are corrected. I will not need to review the paper a second time.

Minor corrections:

Please call the flare SOL2014-09-10 and note its heliographic location and GOES class.

RESPONSE: "The flare SOL2014-09-10T17:45 (X1.6) lasted from 17:21 UT to 18:20 UT (Fig. 2a)"

The assumption about the mass of material needs to be stated in the paper as well as in the Methods section.

RESPONSE: we state how to estimate the mass of material in 2nd paragraph in section F in Result.

Typo "serious" should be "series".

RESPONSE: It has been changed

Typo "motheds" should be "methods".

RESPONSE: It has been changed

Reviewer #4 (Remarks to the Author):

SUMMARY OF THE KEY RESULT: The authors report changes in the rotation of a sunspot around the time of a flare, and assert this reversal is driven by a change in Lorentz force due to the flare.

ORIGINALITY AND INTEREST: This manuscript makes, to my knowledge, the first report of a reversal of rotation associated with a flare. This is probably of interest to a subset of the solar physics community.

That said, the significance of these observations arises primarily from their novelty: beyond being the first such observation, I do not think most solar researchers will be very surprised by it, nor do I think it will lead to any re-evaluation of theories regarding solar flares (i.e., these observations are consistent with current understanding of solar magnetic field dynamics and flares).

Many previous studies (including those cited in the article --- by Anwar et al. and Wang et al. in the 1990s, and more recently by Sudol & Harvey and others) have found flare-associated changes with sunspots and solar magnetic fields. The observations reported here are a type of flare-related magnetic field change not reported before, but the implications of these observations are not, to my mind, substantially different from those discussed in previous reports. The observations discussed in the cited paper by Wang et al. (2014) of flare-related changes in rotation are quite similar (but do not show a reversal of rotation).

Reversals in the direction of rotation of sunspots that were not associated with flares have also been reported before, for instance: <http://adsabs.harvard.edu/abs/2004AAS...204.3716N> <http://adsabs.harvard.edu/abs/2011ApJ...729...95G> These reports should be cited in the paper.

If the authors believe there are more significant implications from their observations, they should highlight these more strongly.

RESPONSE: In this reversion, we compare the result with the torsional oscillations of some sunspots as follows:

"Outside of flares, the torsional oscillations of some sunspots were noticed and the directions of rotation of those sunspots were found to be reversed periodically[41]. The authors reported the period of oscillations as ~ 3.8 day[42]. However, when we traced the rotation of the sunspot reported here using the DAVE4VM velocity field, we found that no torsional oscillations occurred in this sunspot in our studied period and the counter-clockwise rotation of this sunspot last for about two days before the flare."

Moreover, we suggested that the sudden reversal in the rotation of the sunspot implies the magnetic twist transportation from the coronal into the solar interior during the flare.

DATA & METHODOLOGY:

VALIDITY OF APPROACH, QUALITY OF DATA, AND QUALITY OF PRESENTATION:

A terse summary of the paper's approach is: the authors saw a reversal spot rotation; this happened at the same time as a flare; the authors assert that this correlation implies causation, i.e., that the flare caused the change in rotation. While this interpretation is very plausible to me (and in fact I suspect it is correct), the causal link has not

been proven. (Nor could it be, without additional information about the magnetic field's structure and evolution in both the solar interior and atmosphere that cannot be obtained with present observational capabilities.) So the authors should note that while a causal link is physically plausible, such a link is an unproven (but clearly favored) hypothesis. The authors should clearly describe any alternative hypothesis of which they are aware. Near the top of the right column of p.2, the text states flare-associated changes to the photospheric field are "more likely" --- but more likely than what? Perhaps they mean something

like, "Rapid changes in the photospheric field and motions at the photosphere during the flare are more likely driven by the flare than by convective flows, since the photospheric convective turnover time at this length scale is much longer than the timescale of the changes." Note that I substituted "convective timescale" for "Alfvénic timescale" here: if the photospheric field changes are driven by the coronal evolution during the field, these changes must have been mediated by Alfvén waves that propagated down into and then within the photosphere.

The authors' reasoning behind the following statement is unclear: "we obtain the change of the torque during the flare to be..." (p.2, bottom left) and "we can deduce that the torque impulse could cause the observed angular velocity of $20^\circ/\text{hr}$ " (below equation 11). Torque on a point mass is $(r \times F)$, and torque on an extended object requires an integration that includes a factor of r for the moment arm of the force. But the authors only give expressions for F (equation 11). What is the length scale used in computing the torque? In particular, I believe the authors need something like: $\int dx dy (r \times \Delta F)$. This integration should be discussed in the text, assuming it was performed. (The authors could mention that the already cited paper by Wiegmann (2012), on its p. 41, has an expression containing components of the static torque that can be adapted to give the first-order torque arising from changes in B .) How was the moment of inertia estimated? The cited paper by Wang et al. (2014) discusses the choice of rotation axis and estimate of the moment of inertia in detail (on its third page). The angular acceleration mentioned on p.2, $1.4 \text{ degree}/\text{hour}/\text{second}$, multiplied by the 30 seconds mentioned at the end of the same paragraph (and on p.9) would yield a rotation rate of about $40 \text{ deg}/\text{hr}$, which is twice the fitted $20 \text{ deg}/\text{hr}$. How does this discrepancy arise? Is the $1.4 \text{ deg}/\text{hr}/\text{sec}$ figure a peak value, such that the average angular acceleration is half this? I note that based upon the bottom plot of FIG. 1 panel c, the Lorentz torque appears to act for much longer than 30 sec -- perhaps for as long as 300 sec! The authors could address this concern by including a bit more info about their procedures in III. D. on p. 9.

I have no concerns about the quality of the data used: the authors used vector magnetic field and atmospheric imaging measurements from the SDO satellite, which have been employed by many others in studies of solar phenomena. (If magnetogram data from Hindoe/SOT/SP were available, the uncertainties in the magnetic field measurements would likely be reduced, but SP data have limited coverage and very low cadence relative to flare dynamics).

LACK OF USE OF STATISTICS AND TREATMENT OF UNCERTAINTIES: Plots that might have included uncertainties or estimates of variance (e.g., FIG. 2, FIG. 3, FIG. 4) do not include them. Rotation rates (given in degrees / hr) were also presented without uncertainty estimates. This omission is particularly serious in the reported $20\text{-degree}/\text{hr}$ jump associated with the flare. How susceptible is this result to different choices of start/stop times, or different parameters used to fit the ellipse? FIG. 2. panel (a) shows the fitted slope, but what are the uncertainties of this fit?

CONCLUSIONS: The authors' argue that the flare caused the change in sunspot rotation. This is very plausible,

but cannot be conclusively shown with available observations: previous observations of changes in sunspot rotation that were not associated with flares imply that forces arising from within the solar interior could have altered the the spot's rotation, independent of the flare. Accordingly, the authors should rephrase passages of the text that suggest the link has been proven to note the possibility (however remote) that some other causal mechanism is responsible.

To summarize the most significant changes that must be made:

(a) Include uncertainties (for data, fits to data, or quantities derived from data -- like helicity flux), or variances (for things like the average lengths of field lines, or their average twists), in all measured or inferred quantities.

RESPONSE: To estimate the uncertainties, we obtain 50 sets of vector data. In each set, we add a Gaussian noise to three components of the vector magnetic field. The width of the Gaussian function for each pixel is taken as the noise level of the vector magnetic field. We compute the mean field strength, velocity field, helicity flux, and the average lengths and twist of field lines based on each set of vector data. The root mean squares of the results are regarded as the uncertainties in these measurements.

We find that all of the contour lines at contour levels of data number varying between 13000 and 17000 could match well the umbra-penumbral boundary of the sunspot on the intensity images taken around the time of the flare. Supplementary Movie 1 shows the evolution of ellipses fitting the 10 different data-number contour lines. The selected data-numbers range from 13000 to 17000 and space apart from one another at regular interval of 500. Accordingly, we regard the root mean squares of the orientation of the best-fit ellipse at each moment as the uncertainty.

(b) Mention that the observed temporal correlation between the flare and change in spot rotation does not prove causation. The authors certainly can state that they favor the flare-as-cause hypothesis. (I do.)

RESPONSE: we mention the flare-as-cause hypothesis in the First paragraph in Discussion as follows:

"Rapid changes in the photospheric field and motions at the photosphere during the flare are more likely driven by the flare than by convective flows, since the photospheric convective turnover time at this length scale is much longer than the timescale of the changes. Therefore, the impulsive rotation of the sunspot and the rapid change in the magnetic field are more possibly caused by a dynamic process respond to the coronal field reconfiguration during the flare."

(c) Include more description of how the magnetic torque on the sunspot was estimated.

RESPONSE: in this revision, we give an estimate for the length of the underlying magnetic flux coupled with the angular momentum. Taking advantage of the 45-seconds cadence intensity images, we note that the spin motion of the sunspot was accelerated within 96 ± 20 seconds, from 17:29 UT to 17:31 UT (see 3rd paragraph in section D in Result). We adopt the Model S to estimate the density below the photosphere, and then we deduced that the torsional Alfvén pulse could propagate to a depth of about 0.5 Mm below the photosphere in the period of 2 minutes. Thus, the Lorentz torque could apply on the underlying magnetic flux with the depth of 0.5 Mm. The flux tube involved in the reverse rotation would then have the moment of inertia of order of $2 \times 10^{30} \text{ N m}^2$.

This indicates that the sunspot could be accelerated to the maximum angular velocity of $\sim 30^\circ\text{hr}^{-1}$ if the sunspot was accelerated by the Lorentz torque impulse during the entire 2 minutes period. (also see section F in Result)

Beyond these specific issues, more minor scientific issues are:

(d) At the end of the first paragraph on p.1, I am surprised by the authors' outward-biased view: they mention "the role the spinning motion can play in injecting helicity into the corona during a flare." Elsewhere in the paper, however, the authors argue that the changes in photospheric field and forces/torques are driven by the flare -- which originates in the corona. Since the flare is very short compared to timescales of variation of the photospheric field outside of the flaring time interval, I expect spinning motion driven from the interior plays almost no role in helicity transfer during the flare. Because helicity can cross the photosphere in either direction, the text should reflect this: for instance, "the role the spinning motion can play in transferring helicity between the interior and corona (in either direction) during a flare."

RESPONSE: In this revision, "the role the spinning motion can play in injecting helicity into the corona during a flare." is replaced by " the role the spinning motion can play in transferring helicity between the interior and corona (in either direction) during a flare."

(e) p.1, at the first discussion of the flare times (upper-right column), please mention the flare's GOES class. From FIG. 3, it looks $> X1.0$.

RESPONSE: in this revision, when we give the first discussion of the flare, we write that " we characterized the suddenly reversal in the rotation of a part of the sunspot during an X1.6 flare."

(f) Equation (5) in the appendix is incorrect: there should be two integrals, both over the domain S_i (one primed and one unprimed). As written, the units are wrong -- the numerator needs a factor of length² to be a helicity. As a related issue, the domains of the integrations in equation (6) should clearly be labeled as distinct --- e.g., S_j and $S_{i \neq j}$. As with equation (5), equation (8) needs two integrals, both over S_i (one primed and one unprimed).

RESPONSE: equation (5) and Equation (6) in previous version is numbered by Equation (6) and Equation (7) in this reversion, respectively. As pointed out by the reviewer, should be two integrals, both over the domain S_i (one primed and one unprimed). These Equations are rewritten as follows:

$$\dot{\mathbf{H}}_{spin}^i = -\frac{1}{2\pi} \int_{S_i} \int_{S'_i} d^2x d^2x' \hat{\mathbf{n}} \cdot \frac{\mathbf{x} - \mathbf{x}'}{|\mathbf{x} - \mathbf{x}'|^2} \times \{ [V_{\perp t}(\mathbf{x}) - V_{\perp t}(\mathbf{x}')] B_n(\mathbf{x}) B_n(\mathbf{x}') \}, \quad (6)$$

and

$$\dot{\mathbf{H}}_{writhe}^{ij} = -\frac{1}{2\pi} \int_{S_i} \int_{S_j} d^2x d^2x' \hat{\mathbf{n}} \cdot \frac{\mathbf{x} - \mathbf{x}'}{|\mathbf{x} - \mathbf{x}'|^2} \times \{ [V_{\perp t}(\mathbf{x}) - V_{\perp t}(\mathbf{x}')] B_n(\mathbf{x}) B_n(\mathbf{x}') \}. \quad (7)$$

Equation (8) in previous version is rewritten as follow:

$$\bar{\omega}_i = \frac{1}{\Phi_i^2} \int_{S_i} \int_{S'_i} d^2x d^2x' \hat{\mathbf{n}} \cdot \frac{\mathbf{x} - \mathbf{x}'}{|\mathbf{x} - \mathbf{x}'|^2} \times \{ [V_{\perp t}(\mathbf{x}) - V_{\perp t}(\mathbf{x}')] B_n(\mathbf{x}) B_n(\mathbf{x}') \}. \quad (9)$$

(g) Equation (9) is dimensionally flawed: the integral should be normalized by the area integrated in order for the angular velocity to have units of radians per second.

RESPONSE: in previous version, the Equation (9) was written as follows:

$$\bar{\omega}_i = \frac{1}{2} \int_{S_i} d^2x \nabla \times (V_{\perp t}(\mathbf{x})) \quad (9)$$

Here, we rewrite the equation as follows:

$$\bar{\omega}_i = \frac{\int_{S_i} d^2x [(\mathbf{x} - \mathbf{x}_0) \times (V_{\perp t}(\mathbf{x}))]}{\int_{S_i} d^2x}, \quad (10)$$

where \mathbf{x}_0 is where the centroid of region S_i located.

(h) Also, while equation (9) (properly normalized) might yield values close to equation (8), in general the angular velocities computed in these different ways should not be equal: the integrand in equation (8) is weighted by the magnetic flux density, but in (9) it is not. Accordingly, the introductory sentence should be rewritten: "Moreover, an alternative estimate of the mean angular rotation rate within..." (Note also that this sentence should conclude with "as follows:" instead of "as follow")

RESPONSE: we have rewritten the introductory sentence as the reviewer suggested.

(i) Near the end of p.1, why is the difference between the rotation estimates from DAVE4VM (c. 1-3 deg/hr) and ellipse fitting (c. 20 deg/hr) so big? One possibility is that a slight submergence of tilted field (due to inward contraction of the magnetic field as a result of the flare) can lead to a large "apparent footpoint motion" like that discussed by Demoulin & Berger (2003), <http://adsabs.harvard.edu/abs/2003SoPh..215..203D> -- see their Fig. 1, but with submergence instead of emergence. I believe this manuscript's observations might support this apparent footpoint model.

RESPONSE: we note that the rotation estimates from DAVE4VM (c. 1-3 deg/hr) and ellipse fitting (c. 20 deg/hr) is obtained from 12-minutes cadence and 45-seconds cadence data, respectively. "To compare the rotation rate of the fitting ellipses with the mean angular velocity deduced from 12-minutes-cadence vector field, as shown in Fig. 3b, we obtain the rotation rate of ellipse by performing a numerical differentiation to the 12-minutes-average orientations of the ellipse." "Here, a 1σ error indicates the standard deviation of the numerical deviation. It can be seen that, in the phase of the rapid rotation, the change in rotation rate of ellipse is well above the error. In this period, however, the rotation rate of ellipse reached $-10 \pm 3^\circ/\text{hr}$, larger than the average angular velocity of $-4^\circ/\text{hr}$."

In this revision, we discuss the difference between the rotation estimates from DAVE4VM (c. 4 deg/hr) and ellipse fitting (c. 10 deg/hr) as follows:

"It is worth noting that the former denotes the apparent footpoint motion of the umbral. The apparent photospheric motion is partly attributed to the normal plasma velocity. However, the significant vertical flow cannot be detected from DAVE4VM. Another alternative interpretation for the discrepancy would be that the sunspot rotated with slightly inhomogeneous angular velocity and the umbral rotated faster." (see the last paragraph in section D in Result)

(j) Toward the bottom of the left column of p.2 the text states: "could produce upward and vertical torque". Do the authors mean a torque about the vertical axis, in the upward direction? I believe this would yield a counter-clockwise rotation, but the text discusses clockwise rotation. A clockwise rotation would arise from downward torque about the vertical axis.

RESPONSE: After carefully examining the change in Lorentz force, we find that the torque about the vertical axis is indeed downward as the reviewer pointed out. We changed the

sentence that "... upward and vertical torque acting on the sunspot " into "... a torque about the vertical axis in the downward direction. "

(k) In the right col. of p.2, the authors mention field lines in the NLFFF model shortening. I think it is appropriate to repeat the citation of Hudson's (2000) paper on coronal implosion at this point.

RESPONSE: we add the sentence that "Here, similarly, the shortening of the field as well as the increase in the horizontal field of the sunspot refer a unity view, which supports the contraction of the magnetic field lines in an energy-releasing coronal transient event, as the conjecture of a magnetic implosion [8] predicted." (see 4th paragraph in discussion).

(l) p.2: "impulsively, the change"  "impulsively, due to the low Alfvén speed in the interior, the change"

RESPONSE: Here, as reviewer suggested, we changed the sentence that " When α in the corona increases impulsively ..." to the sentence that "Due to the longer timescale of the convective flows in the interior, the change of the interior α should be much lower than in the corona during the period of the flare." (see 3rd paragraph in discussion).

(m) FIG. 1, panels (c) and (d): what is the scale of the V_t , B_t , and dB_t at upper left in each image? This could be stated in the caption, to avoid redoing the figures; but these figures would be more understandable if the scale for these vectors were in the figure itself.

RESPONSE: in this reversion, the scale for these vectors were plotted in the figure itself.

(n) I don't understand the following statement: "Since no significant translational motions was [sic] detected neither in the centroid of P or in the centroid of O in the course of the flare, the impulsive change in $H_{\text{writhe}}^{\text{PO}}$ should also be generated by the abrupt rotation of the sunspot." If the centers of flux of P and O don't move, then they don't wind about each other. So how does this write helicity come about?

RESPONSE: As pointed out by reviewer, the sentence " the impulsive change in $H_{\text{writhe}}^{\text{PO}}$ should also be generated by the abrupt rotation of the sunspot." is wrong.

In this reversion, we point out that "Since no significant translational motions were detected in the centroids of either of "P" or "O" in the course of the flare, the slight change in $H_{\text{writhe}}^{\text{PO}}$ during the flare indicates that the DAVE4VM velocity in the region P do not accurately satisfy a velocity field for a rigid body." (see 4rd paragraph in section B in Result).

(o) For equation (12) in III. E., the first place I am aware of seeing the functional form used was the cited paper by Sudol & Harvey (2005) [6]. I think it should be cited again here.

RESPONSE: it has been cited again here

REFERENCES: While I infer the references were meant to be numbered in the order cited in the paper, they were not always --- see, e.g., [13] then [12] in the manuscript's first paragraph. Citation [18] has no title in the references.

RESPONSE: The modification has been made

A reference to the Sudol & Harvey paper (reference [6] in the manuscript) should be added in the paper's first paragraph, where citations [2,3,9] are already present.

RESPONSE: we added reference [6] in the paper's first paragraph and the sentences are rewritten as follows:

"In particular, rapid and permanent changes of the photospheric magnetic field are found across the time duration of the flare. Consistent with the conjecture of a magnetic implosion [8], the line-of-sight component was observed to be decrease rapidly[6] while the horizontal component was reported to enhance substantially[2, 3, 9]."

In addition to the issues mentioned above, many minor instances of poor grammar, diction, and misspellings must be corrected. (I recommend that the authors run their manuscripts through a spell-checking routine prior to submitting them.) These are enumerated below.

RESPONSE: We have corrected the poor grammar, diction, and misspellings as the reviewer pointed out.

Abstract: "back actions"  "back reactions"

Abstract: "It provides solid evidence..." (unclear referent of "It")  "These observations provide solid evidence..."

Abstract: "a change of Lorentz force exerting on the sunspot"  "a change of Lorentz force exerted on the sunspot"

Abstract: "It support the view"  "These observations support the view"

Abstract: "that the injection of the impulsive helicity flux" is a dynamic process respond to"

 either:

"that helicity is impulsively injected as a dynamic response to" or "that the inferred impulsive helicity flux is a dynamic response to"

Here, "dynamic response to" could be replaced with "dynamic process, in response to", but the latter phrasing uses more words without conveying more information.

p.1: "In particular, the abrupt"  "In particular, abrupt"

p.1: "motion of the magnetic flux"  omit "the"

p.1: "motion of magnetic flux would lead to"  "motion of magnetic flux could lead to" (Not all motions inject helicity. For instance, uniform translation does not.)

p.1: "It was reported in the course of some major flares that "  "It was reported that, in the course of some major flares, "

p.1: "the magnetic helicity was impulsively"  omit "the"

p.1: "across the photosphere and"  insert comma before "and"

p.1: "and the helicity flux tends"  "and that the helicity flux tended"

p.1: "some literatures report"  change "literatures" to "researchers" or "papers"

p.1: "reported the sudden shear-relaxing"  "reported sudden, shear-relaxing"

p.1: "course of the flare and the motions"  "course of flares. These motions..."

p.1: "that the shear-relaxing motion is possible to play"  "that such shear-relaxing motions possibly play"

p.1: "within an isolate"  "within an isolated"

p.1: "is another kind of the photospheric tangential motions." 
"is another kind of photospheric tangential motion."

p.1: "which can be regard as"  "which can be regarded as"

p.1: "a process that transfer magnetic" 
"a process that transfers magnetic"

p.1: "Here, we determined the suddenly reversal" 
"Here, we characterized the sudden reversal"

p.1: "45 seconds cadence"  "45-second cadence"

p.1: "12 minutes cadence vector magnetogram" 
"12 minute cadence vector magnetograms"

p.1: "which is a mature"  "is" to "was"

p.1: "The intensity image (Fig. 1b)"  insert "photospheric" after "The"

p.1: Fig. 1b is discussed before Fig. 1a. Since 1a. shows emission during the flare, I suggest re-ordering the image panels to match their presentation in the text, or re-ordering the text to mention the flare first.

p.1: "into the southern and northern part." 
"into southern and northern parts."

p.2: "It suggests that no..."  "This suggests that no..."

p.2: "It indicates that the impulsive..." 
"This suggests that the impulsive..."

p.2: "we perform B_t in polar"  "we represent B_t in polar" or "we transform B_t into polar"

p.2: "r = 0 locating in"  "r = 0 located in"

p.2: "represented in the radial"  "represented as a"

p.2: "and azimuthal"  "and an azimuthal"

p.2: "changes in B_t was"  change "was" to "were"

p.2: "change of horizontal Lorentz force change acting"  "change in horizontal Lorentz force acting" (omit 2nd change)

p.2: "could produce upward and vertical torque" 

p.2: "which is consistence"  "which is consistent"

p.2: "Using the centroid of the sunspot as the pivot"  "With the centroid of the sunspot as the axis"

p.2: "flare amounts to be"  "flare to be"

p.2: "order higher"  "order of magnitude"

p.2: "It indicates that the sunspot could accelerated to..."  "This indicates that the sunspot could accelerate to..." (or could be accelerated to)

p.2: "lasted for 30 second."  "lasted for 30 seconds."

p.2: "on the photosphere at the flare time"  either "at" or "of" or "in" the photosphere, and "during the flare", since the flare lasts for a finite interval of time

p.2: "a impulsive"  "an impulsive"

p.2: "that are originated from P evolve to have higher"  "that originate from P evolve toward higher"

p.2: "fields is possible to be resulted"  "fields possibly resulted"

p.2: "it gives"  "the models suggest"

p.2: "field lines even if"  "field lines, even if"

p.2: "twist of the field line is"  "twist of the field is"

p.2: "The rotation of the sunspot"  "Outside of flares, the rotation of sunspots"

p.2: "photosphere and then results"  "photosphere, which results"

p.2: "force exerting to the sunspot."  "force exerted on the sunspot."

FIG. 1a. caption: "AIA image at"  "AIA images at"

FIG. 1a. caption: "chromopheres"  "chromosphere"

FIG. 1a. caption: "the flaring loop"  "flaring loops"

FIG. 1b. caption: "intensity image"  "intensity images"

FIG. 1b. caption: "left panels"  "left panel"

FIG. 1b. caption: "close-up view"  "close-up views"

FIG. 1b. caption: "refer to the contour with the contour level of 13000"  "refer to the 13000-data-number contour level." (I assume the units of 13000 are DN.)

FIG. 1b. caption: "which outline"  "which outlines"

FIG. 1b. caption: "by best fitting"  "by fitting"

FIG. 1c. caption: "The image is the"  "The background grayscale image shows the"

FIG. 1c. caption: "vertical field with"  "vertical field, with"

FIG. 1c. caption: "On the right two panels, the close-up view"  "In the right two panels, close-up views"

FIG. 1d. caption: "The image is the vertical field same as in c, on which the superimposed blue arrows refer to the tangential field vectors, while the red arrows on the middle panels refer to changes of the tangential field and the red arrows on the right panel refer to the changes of the tangential Lorentz force."

"As in c, the background grayscale shows the vertical field. In the middle panel, the superimposed blue arrows refer to the tangential field vectors, while the red arrows on the middle panels refer to changes of the tangential field. In the right panel, the red arrows refer to the changes in the tangential Lorentz force."

FIG. 1d. caption: "On each panels, the blue curve outlines a region labeled by P, which matches a flux concentration on the total field images and is identified by a named Yet Another Feature Tracking Algorithm (YAFTA)."

"The blue curve in each panel outlines the region that we label P, which corresponds to a flux concentration (in images of the total magnetic field) identified with the Yet Another Feature Tracking Algorithm (YAFTA) package."

[So this feature was identified in images of |B|, not B_z, correct?]

FIG. 2a. caption: "kinematic parameter"  "kinematic parameters"

FIG. 2a. caption: "is best-fit"  "best fits"

FIG. 2a. caption: "dash line refers"  "dashed lines refer"

FIG. 2a. caption: "The positive (resp. negative)"  "Positive (resp. negative)"

FIG. 2a. caption: "the counter-clockwise (resp. clockwise)"  "counter-clockwise (resp. clockwise)"

FIG. 2b and 2c captions: these were interchanged; please switch them.

FIG. 2b and 2c captions: capitalize start of first sentence, "the"  "The"

FIG. 2b. caption: "exerting to"  "exerted on"

Fig. 3. caption: The labels for each panel are incorrect. Panel (a), the GOES light curve, is not described at all.

Fig. 3a caption: The text after (a) in the current caption describes (b). "14-hours averaged"  "14-hour averaged"

Fig. 3b caption: This caption describes (c). Capitalize first word, "the".

Fig. 3c caption: This caption describes (d). Capitalize first word, "the".

Fig. 3c caption: "over the region as the FOV of panel (a)"  "over the same FOV as panel (b)"

Fig. 3d caption: This refers to the panel labeled (e). Capitalize the first word, "the." Later, "the blue line refer to"  "The blue line refers to"

Fig. 3e caption: I think this refers to a panel that does not exist.

p. 8, III. (section title): "MOTHEDS"  "Methods"

p. 8, III.A., "HMI takes the full-disk"  omit "the"

p. 8, III.A., "[32] and the remaining"  "[32]. The remaining"

p. 8, III.A., "the mean"  "a mean"

p. 8, III.B., "is ellipse"  "is elliptical"

p. 8, III.B., "of the ellipse that best-fit it"  "of the best-fit ellipse"

p. 8, III.B., "it starts"  "it started"

p. 8, III.B., "several hours after"  "until several hours after"

p. 8, III.B., "It indicates"  "This indicates"

p. 8, III.C. (subsection title): "Moetheds of helcity" (two misspellings)  "Methods for helicity fluxes"

p. 8, III.C., "donates the vector"  "denotes the vector"

p. 8, III.C., "components of velocity V_{\perp} , the velocity"  "components of V_{\perp} , the velocity"

p. 8, III.C., "to the magnetic field lines."  "to the magnetic field."

p. 8, III.C., "from the the twisted"  omit "the the", so "from twisted"

p. 8, III.C., "by the tangential"  "by tangential"

p. 8, III.C., "fluxes are mainly contributed by the isolate Photospheric"  "fluxes mainly consist of isolated photospheric"

p. 8, III.C., "contributed by the horizontal motions of the flux elements"  "due to horizontal motions of flux elements"

p. 8, III.C., "from the weak flux outside of the flux concentrations"  "from weak fields outside flux concentrations"

p. 8, III.C., "can be negligible."  "can be neglected."

p. 8, III.C., interchanged reference:

"the spin term (H_{sp}) and writhe term (H_{wh}), which refers to the contribution from relative proper motions of photospheric flux elements about one another and from internal spinning motions within each element, respectively."

 switch H_{sp} and H_{wh} to match descriptions,

"the writhe term (H_{wh}) and spin term (H_{sp}), which refers to the contribution from relative proper motions of photospheric flux elements about one another and from internal spinning motions within each element, respectively."

p. 9, III.C., "we define the region as "P", which is tracked by a named Yet Another Feature Tracking Algorithm (YAFTA) [14] and encloses a flux concentration covering the rotational part of the"

"we used the Yet Another Feature Tracking Algorithm (YAFTA) package [14] to define and track the region labelled "P". This region encloses a flux concentration that comprises the rotational part of the"

p. 9, III.C., "sunspot, while the region outside the region 'P' is" 
"sunspot. The region outside 'P' is"

p. 9, III.C., "the equation (4)"  omit "the"

p. 9, III.C., "motions was detected neither in the centroid of P or in the centroid of O"  "motions were detected in the centroids of either of P or O"

p. 9, III.C., "Indicated"  "As indicated"

p. 9, III.C., "but the reversal"  omit "but"

p. 9, III.D., "the Lorentz force are"  "are" to "is"

p. 9, III.D., "pressure gradient"  "pressure gradients"

p. 9, III.D., "and then produce"  "which could then produce"

p. 9, III.D., "We makes the simple assumption"  "We make the simplifying assumption"

p. 9, III.D., "time delta t, over which"  omit comma

p. 10, III.F., "is used to be the"  "is used as"

p. 10, III.F., "twist of magnetic"  "twist of a magnetic"

Once again, thank you very much for your comments and suggestions.

Yours sincerely,

Bi

Reviewer #1 (Remarks to the Author):

I thank the authors for the replies and changes to the manuscript.

Supplementary movie 1 was very helpful. The southern part of the sunspot intensity pattern clearly moves to the left (eastward) during the movie. Both the east and west edges of this portion of the sunspot move eastward, consistent with an impulsive clockwise rotation of this part of the sunspot.

In Figure 3a the angular speed deduced by DAVE4VM remains significantly negative for approximately an hour after the flare, whereas the sign of the angular velocity from the ellipse-fitting to intensity maps is undetermined in Figure 3b because of the uncertainties. The minimum at flare time in Figure 3b is much larger than in Figure 3a. These plots are discussed in Section I Subsection D. My guess is that the abrupt change was interpreted in the ellipse fitting as a strictly proper bulk motion, whereas much of the apparent motion at flare time may have instead been signatures of opposite changes in sunspot structure at the east and west penumbrae, e.g., flattening/darkening of the penumbra at the east and steepening/brightening at the west, not interpreted as proper motion by DAVE4VM. Can the authors clarify this point? In any case these plots show more consistency than previously.

In the equation for the moment of inertia in Section I Subsection F, the second r_i should become r_0 and the second \prime should be deleted.

The calculation for the corona-driven angular acceleration in Section I Subsection F does indeed suggest that the impulsive coronal torque might be enough to cause the observed sudden change in angular velocity. However, part of my initial criticism was not addressed: the reaction of the field below the surface to the coronal torque is not taken into account in the calculation. There may be more opposition from the interior to the coronal change than the authors suppose. Unless the Alfvén transit time in the interior of the sunspot is unequivocally much longer than the duration of the corona-driven change, the interior field cannot be eliminated as an active participant in the dynamics. This issue should be discussed in the text of Subsection F and accounted for in the conclusions.

Reviewer #2 (Remarks to the Author):

The manuscript is improved. My only remaining comment is that I think Cameron and Sammis 1999 observed the line-of-sight field, and hence should be cited with the Sudol & Harvey reference (i.e.):
'the line-of-sight component was observed to be decrease rapidly[3,6]'

Reviewer #4 (Remarks to the Author):

The manuscript is much improved. Several passages, however, still contain flawed English, and these will need to be edited.

Issues of grammar aside, I have only a few minor concern's about the manuscript's scientific content. Each of the following points should be simple to correct.

1. Unstated assumption, p.1: After "The clockwise (counter-clockwise) rotation", insert "of a upwardly directed magnetic flux tube" before "transfers helicity of positive..."
2. Unstated noise levels, p.2: "In each experiment, we add a Gaussian noise... The width of the

Gaussian function ... is taken as the noise level of the vector magnetic field." Were noise levels assumed? Or were these adopted from noise estimates made by the HMI Team? What are some typical values of these noise estimates? Please add a sentence or two giving the size of the noise levels, and where you got the noise estimates.

3. Reversed terms, p.2: Where the text says "by the vertical and tangential motion of the magnetic flux, respectively", I believe "vertical" and "tangential" are in the wrong order.

4. Improperly formatted equation, p.4: in the equation " $I = \sum m_i (r_i - (r_i)^2)$ ", there two problems: either a missing right parenthesis or extra left parenthesis; and r_i instead of r_0 .

Reviewer #1 (Remarks to the Author):

I thank the authors for the replies and changes to the manuscript.

Supplementary movie 1 was very helpful. The southern part of the sunspot intensity pattern clearly moves to the left (eastward) during the movie. Both the east and west edges of this portion of the sunspot move eastward, consistent with an impulsive clockwise rotation of this part of the sunspot.

In Figure 3a the angular speed deduced by DAVE4VM remains significantly negative for approximately an hour after the flare, whereas the sign of the angular velocity from the ellipse-fitting to intensity maps is undetermined in Figure 3b because of the uncertainties. The minimum at flare time in Figure 3b is much larger than in Figure 3a. These plots are discussed in Section I Subsection D. My guess is that the abrupt change was interpreted in the ellipse fitting as a strictly proper bulk motion, whereas much of the apparent motion at flare time may have instead been signatures of opposite changes in sunspot structure at the east and west penumbrae, e.g., flattening/darkening of the penumbra at the east and steepening/brightening at the west, not interpreted as proper motion by DAVE4VM. Can the authors clarify this point? In any case these plots show more consistency than previously.

In the equation for the moment of inertia in Section I Subsection F, the second r_i should become r_0 and the second \cdot should be deleted.

The calculation for the corona-driven angular acceleration in Section I Subsection F does indeed suggest that the impulsive coronal torque might be enough to cause the observed sudden change in angular velocity. However, part of my initial criticism was not addressed: the reaction of the field below the surface to the coronal torque is not taken into account in the calculation. There may be more opposition from the interior to the coronal change than the authors suppose. Unless the Alfvén transit time in the interior of the sunspot is unequivocally much longer than the duration of the corona-driven change, the interior field cannot be eliminated as an active participant in the dynamics. This issue should be discussed in the text of Subsection F and accounted for in the conclusions.

Reviewer #2 (Remarks to the Author):

The manuscript is improved. My only remaining comment is that I think Cameron and Sammis 1999 observed the line-of-sight field, and hence should be cited with the Sudol & Harvey reference (i.e.):
'the line-of-sight component was observed to be decrease rapidly[3,6]'

Reviewer #4 (Remarks to the Author):

The manuscript is much improved. Several passages, however, still contain flawed English, and these will need to be edited.

Issues of grammar aside, I have only a few minor concern's about the manuscript's scientific content. Each of the following points should be simple to correct.

1. Unstated assumption, p.1: After "The clockwise (counter-clockwise) rotation", insert "of a upwardly directed magnetic flux tube" before "transfers helicity of positive..."
2. Unstated noise levels, p.2: "In each experiment, we add a Gaussian noise... The width of the

Gaussian function ... is taken as the noise level of the vector magnetic field." Were noise levels assumed? Or were these adopted from noise estimates made by the HMI Team? What are some typical values of these noise estimates? Please add a sentence or two giving the size of the noise levels, and where you got the noise estimates.

3. Reversed terms, p.2: Where the text says "by the vertical and tangential motion of the magnetic flux, respectively", I believe "vertical" and "tangential" are in the wrong order.

4. Improperly formatted equation, p.4: in the equation " $I = \sum m_i (r_i - (r_i)^2)$ ", there two problems: either a missing right parenthesis or extra left parenthesis; and r_i instead of r_0 . Gaussian function ... is taken as the noise level of the vector magnetic field." Were noise levels assumed? Or were these adopted from noise estimates made by the HMI Team? What are some typical values of these noise estimates? Please add a sentence or two giving the size of the noise levels, and where you got the noise estimates.

3. Reversed terms, p.2: Where the text says "by the vertical and tangential motion of the magnetic flux, respectively", I believe "vertical" and "tangential" are in the wrong order.

4. Improperly formatted equation, p.4: in the equation " $I = \sum m_i (r_i - (r_i)^2)$ ", there two problems: either a missing right parenthesis or extra left parenthesis; and r_i instead of r_0 .

Dear Reviewer:

Thank you for your comments concerning our manuscript. Those comments are all valuable and very helpful for revising and improving our paper, as well as the important guiding significance to our researches. We have studied comments carefully and have made correction, which we hope meet with approval.

In the file of "changes_are_marked.pdf", the portions marked in red letters indicate changes as listed in the point-by-point response to the referees' comments, while the portions marked in blue letters denote some poor grammars that are corrected by authors. The main corrections in the paper and the responds to your comments are as following

Reviewers' comments:

Reviewer #1 (Remarks to the Author):

I thank the authors for the replies and changes to the manuscript.

Supplementary movie 1 was very helpful. The southern part of the sunspot intensity pattern clearly moves to the left (eastward) during the movie. Both the east and west edges of this portion of the sunspot move eastward, consistent with an impulsive clockwise rotation of this part of the sunspot.

In Figure 3a the angular speed deduced by DAVE4VM remains significantly negative for approximately an hour after the flare, whereas the sign of the angular velocity from the ellipse-fitting to intensity maps is undetermined in Figure 3b because of the uncertainties. The minimum at flare time in Figure 3b is much larger than in Figure 3a. These plots are discussed in Section I Subsection D. My guess is that the abrupt change was interpreted in the ellipse fitting as a strictly proper bulk motion, whereas much of the apparent motion at flare time may have instead been signatures of opposite changes in sunspot structure at the east and west penumbrae, e.g., flattening/darkening of the penumbra at the east and steepening/brightening at the west, not interpreted as proper motion by DAVE4VM. Can the authors clarify this point? In any case these plots show more consistency than previously.

RESPONSE: As the Reviewer noted, we state in the first paragraph of Subsection F that "Supplementary Movie 1 created from the intensity images shows that both the southeast and southwest edges of the sunspot move eastward and that the light bridge, located along the northeast edge of the southern part of the sunspot, moves westward. This is consistent with an impulsive clockwise rotation of this part of the sunspot." To clearly show the motion of the light bridge in the animation, we plot a blue line in a fixed position to denote the location of the light bridge at the beginning of the animation.

In the last paragraph of Subsection F, we suggest that "Moreover, the abrupt change in the ellipse fitting could not be completely attributed to a strictly proper bulk motion of the sunspot, but could be partly attributed to signatures of opposite changes in sunspot structure at the penumbrae, e.g., flattening of the penumbra at the southeast or steepening at the southwest."

In the equation for the moment of inertia in Section I Subsection F, the second r_i should become r_0 and the second '(' should be deleted.

RESPONSE: It has been changed.

The calculation for the corona-driven angular acceleration in Section I Subsection F does indeed suggest that the impulsive coronal torque might be enough to cause the observed sudden change in angular velocity. However, part of my initial criticism was not addressed: the reaction of the field below the surface to the coronal torque is not taken into account in the calculation. There may be more opposition from the interior to the coronal change than the authors suppose. Unless the Alfvén transit time in the interior of the sunspot is unequivocally much longer than the duration of the corona-driven change, the interior field cannot be eliminated as an active participant in the dynamics. This issue should be discussed in the text of Subsection F and accounted for in the conclusions.

RESPONSE: In the version, this issue is discussed in the text of Subsection F and accounted for in the conclusions. We suggested that there may be opposition from the interior to the coronal change. Moreover, we suggest that the reaction of the field below the surface may produce an opposite torque to balance the magnetic torque acted on the photosphere.

In the third paragraph of Subsection F, we add the sentences that “If the photospheric field changes impulsively, so that the magnetic field below the photosphere does not have time to respond, the changes in the Lorentz force applied to the photosphere would cause a net rotational torque exerted on the sunspot and then produce an angular acceleration of the sunspot.”

In the Discussion, we add the sentences that “Moreover, the Lorentz torque could be sufficient to accelerate the reversal of rotation in the sunspot on a time scale of 2 minutes. It must be noted that the reaction of the field below the surface to the coronal torque is not taken into account in the calculation. There may be opposition from the interior to balance the change in the field on the photosphere. However, the change in the interior of the sunspot responding to the coronal field reconfiguration may lag behind the change in the photosphere.”

Reviewer #2 (Remarks to the Author):

The manuscript is improved. My only remaining comment is that I think Cameron and Sammis 1999 observed the line-of-sight field, and hence should be cited with the Sudol & Harvey reference (i.e.):

'the line-of-sight component was observed to be decrease rapidly[3,6]

RESPONSE: It has been changed.

Reviewer #4 (Remarks to the Author):

The manuscript is much improved. Several passages, however, still contain flawed English, and these will need to be edited.

Issues of grammar aside, I have only a few minor concern's about the manuscript's scientific content. Each of the following points should be simple to correct.

1. Unstated assumption, p.1: After "The clockwise (counter-clockwise) rotation", insert "of a upwardly directed magnetic flux tube" before "transfers helicity of positive..."

RESPONSE: It has been changed

2. Unstated noise levels, p.2: "In each experiment, we add a Gaussian noise... The width of the Gaussian function ... is taken as the noise level of the vector magnetic field." Were noise levels assumed? Or were these adopted from noise estimates made by the HMI Team? What are some typical values of these noise estimates? Please add a sentence or two giving the size of the noise levels, and where you got the noise estimates.

RESPONSE: in the first paragraph of Subsection B, we add the sentences that "The width of the Gaussian function for each pixel is taken as the noise level of the vector magnetic field, which is estimated based on the inversion code and provided by the HMI team[28]. The uncertainty in the vertical and tangential field is about 15 G and 50 G at each pixel, respectively."

3. Reversed terms, p.2: Where the text says "by the vertical and tangential motion of the magnetic flux, respectively", I believe "vertical" and "tangential" are in the wrong order.

RESPONSE: It has been changed

4. Improperly formatted equation, p.4: in the equation " $I = \sum m_i (r_i - (r_i)^2)$ ", there two problems: either a missing right parenthesis or extra left parenthesis; and r_i instead of r_0 .

RESPONSE: It has been changed

Once again, thank you very much for your comments and suggestions.

Yours sincerely,

Bi